# Creating a Causally Grounded Rating Method for Assessing the Robustness of AI Models for Time-Series Forecasting

## Abstract

AI models, including both time-series-specific and general-purpose Foundation Models (FMs), have demonstrated strong potential in time-series forecasting across sectors like finance. However, these models are highly sensitive to input perturbations, which can lead to prediction errors and undermine trust among stakeholders, including investors and analysts. To address this challenge, we propose a causally grounded rating framework to systematically evaluate model robustness by analyzing statistical and confounding biases under various noisy and erroneous input scenarios. Our framework is applied to a large-scale experimental setup involving stock price data from six companies and evaluates both uni-modal and multi-modal models, including Vision Transformer-based (ViT) models and FMs. We introduce six types of input perturbations and twelve data distributions to assess model performance. Results indicate that multi-modal and time-series-specific FMs demonstrate greater robustness and accuracy compared to general-purpose models. Further, to validate our framework's usability, we conduct a user study showcasing time-series models' prediction errors along with our computed ratings. The study confirms that our ratings reduce the difficulty for users in comparing the robustness of different models. Our findings can help stakeholders understand model behaviors in terms of robustness and accuracy for better decision-making even without access to the model weights and training data, i.e., black-box settings.

## 1 Introduction

Time series (TS) forecasting uses historical data indexed by time to predict future values. This task finds wide applicability in industry in domains like finance, healthcare, manufacturing, and weather. Although well-studied, the TS forecasting has seen recent advancements with new AI-based approaches including gradient boosting, deep learning, transformers and Foundation Models (FMs) trained on uni-modal numerical data as well as multi-modal data vying for state-of-the-art performance (Elsayed et al. (2021); Jin et al. (2023)).

However, having good performance is no guarantee that users will trust a method or model and use it. In particular, users care about the model's robustness to noisy data and perturbations, as erroneous predictions can have far-reaching impact on stakeholders. The perturbations may have been caused unintentionally by an actor or intentionally by an adversary, but regardless, the users expect robust and consistent performance. To manage user trust, a promising idea is of third-party assessment of models and ratings (automated certifications), which can help users make informed decisions, even without access to the method's code or model's training data using both statistical (Srivastava & Rossi (2018; 2020); Srivastava et al. (2024)) and causality-based methods (Lakkaraju et al. (2023; 2024)).

In this context, our contributions are that we:

1. We propose a rating workflow for rating time-series forecasting models (TSFM) for robustness, extending previous rating methods (Srivastava & Rossi (2018; 2020); Srivastava et al. (2024); Lakkaraju et al. (2023; 2024)). This workflow supports a new use case of model selection for time-series forecasting based on robustness and forecasting accuracy.

2. We define six input perturbations for evaluation: Input-specific Perturbations (IP) (2), Semantic Perturbations (SP) (2), Syntactic Perturbation (SyP) (1), and Composite Perturbation (CP) (1), which includes a joint evaluation with an image-based sentiment analysis system. We evaluate these models using one year of stock price data from six leading companies across three industries.

3. Our rating method introduces two novel causality-based metrics alongside established ones, generating ratings to compare three baseline models along with 8 time-series forecasting models (which include 6 foundation models and 2 ViT-num-spec models) on forecasting accuracy and robustness.

4. We conduct a user study to evaluate the usability of ratings to interpret model behavior. The study confirms that our ratings reduce the difficulty for users in comparing the robustness of different models.

5. In addition to the core research questions outlined in Section 3, through our experiments, we also answer additional research questions (ARQs) in Section 5.3:

   - ARQ1: How do different model types compare in performance? (Foundation Vs. Non-foundation models)
   - ARQ2: Does multi-modality improve the performance of TSFM? (uni-modal vs. multi-modal)
   - ARQ3: How does the architecture of FMs influence performance? (time series-specific vs. general-purpose and encoder-only vs. decoder-only vs. encoder-decoder)

## 2 Related Work

We now contextualize our work with related literature so that our contributions are highlighted. We cover Transformer-based TSFM models, perturbations in finance domain, robustness testing of TSFM, causal analysis of TSFM, and rating AI models.

### 2.1 Time-series Forecasting with Transformer-based Models

Transformer architectures have gained significant traction in time-series forecasting due to their effectiveness in capturing long-term temporal patterns and handling variable input lengths. (Lu et al. (2022)) shows that transformers pre-trained on text data can solve sequence modeling tasks in other modalities paving the way for leveraging language pre-trained transformers for time series analysis. (Zhou et al. (2023)) and (Jin et al. (2023)) further illustrate the versatility and robustness of fine-tuned language pre-trained transformers for diverse time series tasks. (Cheng et al. (2022)) introduces a multi-modal graph neural network for learning from multi-modal inputs. These works typically use data from various sources. In contrast, (Zeng et al. (2023)) introduces a vision transformer using time-frequency spectrograms to transform numerical data into a multi-modal form, showing benefits in both time and frequency domains. We extend this work by evaluating two variations of the ViT-num-spec model from (Zeng et al. (2023)). These works typically use data from multiple sources to support multi-modality or operate solely on single-modality data such as raw numerical sequences. In contrast, (Zeng et al. (2023)) transforms simple numerical time-series data into a multi-modal representation using time-frequency spectrograms, enabling a vision transformer to jointly learn from both time intensities and frequency spectrograms. We extend this work by evaluating two variants of the ViT-num-spec model proposed in (Zeng et al. (2023)).

Building on the strengths of transformers, recent developments have introduced FMs that extend these capabilities to multi-task settings, enabling zero-shot forecasting and broader generalization across other domains including time-series. Recent studies have reprogrammed LLMs for time series tasks through parameter-efficient fine-tuning and tokenization strategies (Zhou et al. (2023); Gruver et al. (2023); Jin et al. (2023); Cao et al. (2023); Ekambaram et al. (2024)). (Ansari et al. (2024)) and (Woo et al. (2024)) have improved forecasting accuracy and model generalization, while (Rasul et al. (2023)) and (Das et al. (2023)) have explored new tokenization strategies and fine-tuning methods. (Yu et al. (2023)) leverages historical stock prices, company metadata, and economic news for explainable time series forecasting with LLMs. (Garza & Mergenthaler-Canseco (2023)) and (Ekambaram et al. (2024)) developed lightweight models for real-time applications, and (Talukder et al. (2024)) integrated multiple temporal patterns to improve

precision. FMs trained from scratch, like (Gruver et al. (2023)), achieved SOTA in zero-shot forecasting, with (Cao et al. (2023)) and (Goswami et al. (2024)) further improving model performance. In our experiments, we select Gemini-V and Phi-3 as the General Purpose FMs and Chronos and MOMENT as Time-series-specific FMs due to their SOTA performance in their respective categories.

## 2.2 Perturbations in Finance Domain

TS data is commonly stored in spreadsheets and databases, which are prone to changes due to acts of omission (e.g., negligence, data-entry errors) or commission (e.g., adversarial attacks, sabotage). Omission errors are most common (Panko & Halverson (1996)). Tools like Microsoft Excel and Google Sheets are widely used for data collection and analysis, allowing end-user programming (Birch et al. (2018)). However, over 90% of spreadsheets contain errors due to issues like incorrect formulae, leading to multi-billion dollar losses (Pak-Lok POON & TANG (2024)). Adversarial attacks are also increasing in data stores and AI models for tasks like forecasting. Malicious agents target ML models in financial institutions for financial gain, exploiting the limited robustness of deep-learning architectures commonly used in NLP. (Karim et al. (2019)) explore both black-box and white-box attacks in time-series forecasting task. (Oregi et al. (2018)) revealed the vulnerability of distance-based classifiers. (Rathore et al. (2020)) examined various adversarial attacks on time series classifiers. TSFool (Li et al. (2022)) introduced a multi-objective black-box attack to craft imperceptible adversarial time series to fool RNN classifiers. (Nehemya et al. (2021)) highlights the vulnerability of algorithmic trading systems to real-time adversarial attacks using imperceptible perturbations, highlighting the need for mitigation strategies. (Fursov et al. (2021)) examines adversarial attacks on deep-learning models in financial transaction records, revealing significant vulnerabilities. However, these works do not explore multi-modal models or multi-modal attacks. Our work addresses several perturbations applicable to uni-modal and multi-modal models, using both common data errors and attacks to measure the robustness of TSFM.

## 2.3 Robustness Testing of Time-series Forecasting Models

(Gallagher et al. (2022)) examines the impact of different attacks on the performance of CNN model used for time series classification. (Pialla et al. (2023)) introduces a stealthier attack using the Smooth Gradient Method (SGM) for time series and measures the effectiveness of the attack. While (Pialla et al. (2023)) focuses on measuring the smoothness of the attacks, our work quantifies their impact on the models in addition to the biases they create in models' predictions. (Govindarajulu et al. (2023)) adapt attacks from the computer vision domain to create targeted adversarial attacks. They examine the impact of the proposed targeted attacks versus untargeted attacks using statistical measures. All these works measure the models' performances under perturbations using statistical methods but do not measure the isolated impact of perturbations which is only possible through causal analysis. Furthermore, they do not consider any transformer-based or multi-modal models for evaluation.

## 2.4 Causal Analysis in Time-series Forecasting

(Moraffah et al. (2021)) provides a review of the approaches used to compute treatment effects and also discusses causal discovery methods along with commonly used evaluation metrics and datasets. As the perturbations we introduce in this paper do not occur at the same timestep in each sample, the effect we are measuring can be considered as time-varying perturbation effect which is more complex to measure compared to time-invariant treatment effects. (Robins et al. (1999)) measures the causal effect of one such time-varying exposure. However, they only consider binary outcomes. In our work, we deal with continuous outcomes and analyze the treatment effects of multiple treatments in the presence of confounders that are of interest in the time-forecasting domain.

## 2.5 Rating AI Models

Several works have assessed and rated AI models for trustworthiness from a third-party perspective without access to training data. (Srivastava & Rossi (2020)) proposed a method to rate AI models for bias, specif-

ically targeting gender bias in machine translators (Srivastava & Rossi (2018)), and used visualizations to communicate these ratings (Bernagozzi et al. (2021a)). They conducted user studies on trust perception through visualizations (Bernagozzi et al. (2021b)), but these lacked causal interpretation. (Lakkaraju et al. (2024)) introduced a causal analysis approach to rate bias in sentiment analysis systems, extending it to assess their impact when used with translators (Lakkaraju et al. (2023)). We extend this method to rate TSFM for robustness against perturbations. Causal analysis offers advantages over statistical analysis by determining accountability, aligning with humanistic values, and quantifying the direct influence of various attributes on forecasting accuracy.

## 3 Problem

### 3.1 Preliminaries

**Time Series Forecasting** Let the time series be represented by $\{x_{t-n+1}, x_{t-n+2}, ...., x_t, x_{t+1}, ..., x_{t+d}\}$, where each $x_{t-n+i}$ represents a value in time series, where $n$ is called the sliding window size and $d$ is the number of future values the model predicts. Let $X_t = \{x_{t-n+1}, x_{t-n+2}, ...., x_t\}$, and $\hat{Y}_t = \{\hat{x}_{t+1}, \hat{x}_{t+2}, ...., \hat{x}_{t+d}\}$, where $\hat{Y}_t = f(X_t; \theta)$ for uni-modal TSFM, and in the case of multi-modal TSFM, $X_t$ includes a combination of numerical time-series values, time-series line plots, and time-frequency spectrograms. The function $f$ represents pre-trained TSFM with parameter $\theta$ that predicts the values for the next 'd' timesteps based on the values at previous $n$ timesteps. Let $Y_t$ denote the true values for the next 'd' timesteps. Let $S$ be the set of TSFM we want to rate. Let $R_t$ be the residual for the sliding window [t + 1, t + d] and is computed by $(\hat{Y}_t - Y_t)$ at each timestep. Our rating method aims to highlight the worst-case scenario for the model. Therefore, we consider the maximum residual, denoted as $R_t^{max}$.

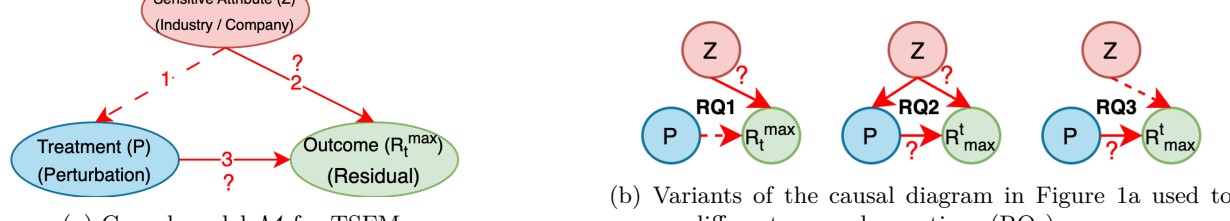

(a) Causal model $\mathcal{M}$ for TSFM.

(b) Variants of the causal diagram in Figure 1a used to answer different research questions (RQs).

Figure 1: (a) Causal model $\mathcal{M}$ for TSFM. The validity of link '1' depends on the data distribution $(P|Z)$, while the validity of the links '2' and '3' are tested in our experiments. (b) Variants of $\mathcal{M}$ used to answer different research questions (RQs).

**Causal Model** The causal model $\mathcal{M}$, is shown in Figure 1a. Arrowheads indicate the causal direction from cause to effect. If *Sensitive Attribute* ($Z$) is a common cause for both *Perturbation* ($P$) and *Residual* ($R_t^{max}$), it introduces a spurious correlation between $P$ and $R_t^{max}$, known as the confounding effect, making $Z$ the confounder. The path from *Perturbation* to *Residual* through the confounder is called a *backdoor path* and is undesirable. Various backdoor adjustment techniques can remove the confounding effect (Xu & Gretton (2022); Fang et al. (2024); Liu et al. (2021)). The deconfounded distribution, after adjustment, is represented as $(R_t^{max}|do(P))$. The 'do(.)' operator in causal inference denotes an intervention to measure the causal effect of $P$ on the $R_t^{max}$. Solid red arrows with '?' in Figure 1a denote the causal links tested in our experiments, while the dotted red arrow represents a potential causal link, depending on the distribution $(P|Z)$ across different values of $Z$.

### 3.2 Problem Formulation

We aim to answer the following research questions (RQs) (with causal diagrams in Fig 1b) through our causal analysis when different perturbations denoted by $P = \{0, 1, 2, 3\}$ (or simply $P0, P1, ...$) are applied to the input given to the set of TSFM $S$:

**RQ1: Does $Z$ affect $R_t^{max}$, even though $Z$ has no effect on $P$?** That is, if perturbations are independent of the sensitive attribute, can the sensitive attribute still affect the model outcome, leading to statistical bias (i.e., lack of fairness)? *(Perturbations are distributed uniformly with respect to $Z$; see Fig. 1b, left).*

**RQ2: Does $Z$ affect the relationship between $P$ and $R_t^{max}$ when $Z$ has an effect on $P$?** That is, if the applied perturbations depend on the value of the sensitive attribute, would the sensitive attribute add a spurious (false) correlation between the perturbation and the outcome of a model leading to confounding bias? *(Perturbations vary systematically with $Z$; see Fig. 1b, middle).*

**RQ3: Does $P$ affect $R_t^{max}$ when $Z$ may have an effect on $R_t^{max}$?** That is, what is the impact of the perturbation on the outcome of a model when the sensitive attribute may still have an effect on the outcome of a model? *(Same setup as RQ2, but focused on estimating the perturbation's effect; see Fig. 1b, right).*

**RQ4: Does $P$ affect the accuracy of $S$?** That is, do the perturbations affect the performance of the models' accuracy? Causal analysis is not required to answer this question as we only need to compute appropriate accuracy metrics to assess how robust a model is against different perturbations.

## 4 Solution Approach

Our solution approach comprises the following components: (1) A set of six perturbations applied selectively to numerical time series data, time-frequency spectrograms, time-series intensities, and time-series line plots. (2) Robustness evaluation metrics: WRS, APE, and PIE %, and forecasting accuracy metrics, each used separately to compute the ratings. (3) A structured workflow that maps data to predictions and predictions to ratings, enabling the evaluation of TSFM using both uni-modal (numerical) and multi-modal (line plots, time-frequency, and time-intensity data combined with numerical data) inputs.

### 4.1 Perturbations

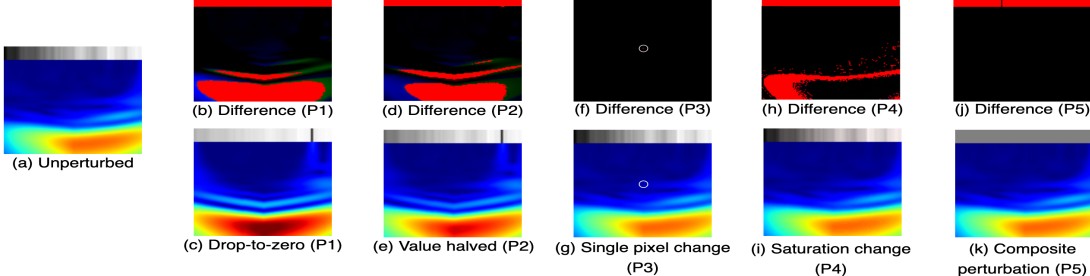

(a) Unperturbed

(b) Difference (P1)    (d) Difference (P2)    (f) Difference (P3)    (h) Difference (P4)    (j) Difference (P5)

(c) Drop-to-zero (P1)    (e) Value halved (P2)    (g) Single pixel change (P3)    (i) Saturation change (P4)    (k) Composite perturbation (P5)

Figure 2: Time-frequency spectrogram representations of time-series data and their perturbed variants. The original (unperturbed) spectrogram is shown alongside images generated after applying different perturbations: semantic perturbations (P1: drop-to-zero, P2: value halved), syntactic perturbation (P3: missing values), input-specific perturbations applied to spectrogram images (P4: single-pixel change, P5: saturation change), and a composite perturbation (P6). Each perturbed image is accompanied by a corresponding difference image (highlighted in red) indicating the regions modified by the perturbation. In (f) and (g), the single-pixel modification is emphasized with a white circle.

We define six perturbations: two semantic (SP), one syntactic (SyP), two input-specific (IP), and one composite (CP) (Figure 2) inspired by real-world applications in unintended scenarios to assess the robustness and accuracy of TSFM.

**a) Semantic Perturbation (SP):** Semantic perturbations are alterations made to data that change its meaning while preserving the overall context. For example, in time-series forecasting, a stock's value might change drastically due to errors in data entry, or it might fluctuate due to some market-specific catalyst that affects certain companies. Under SP, we consider two perturbations:

**1. Drop-to-zero (P1)**: It is inspired by common data entry errors (Ley et al. (2019)). Every $n^{th}$ value in the original stock price data is set to zero. Sampling the time series with a sliding window of size $n$ ensures each sample contains a zero.

**2. Value halved perturbation (P2)**: Every $n^{th}$ value in the original stock price data is reduced to half of its value. This perturbation simulates periodic adjustments, possibly reflecting events like stock splits or dividend payments.

**b) Syntactic Perturbation (SyP):** SyPs modify the structure of the data without altering its fundamental meaning. We consider one such perturbation.

**1. Missing values perturbation (P3)** Every $n^{th}$ value in the original stock price data is converted to a null value, simulating real-world missing data points in financial datasets due to incomplete transmissions.

**c) Input-specific Perturbation (IP)**: Input-specific perturbations are alterations specific to the mode - features and context of the data being used. In time-series forecasting, altering some pixels (e.g., changing their color) in a time-frequency spectrogram image is one example. Under IP, we consider two different perturbations:

**1. Single pixel change (P4):** In (Su et al. (2019)), the authors modified a single pixel in each of the test images. With this approach, they fooled three types of DNNs trained on the CIFAR dataset. In our work, we alter the center pixel of each multi-modal input to black based on the intuition that small and consistent change to the images can significantly alter the models' predictions.

**2. Saturation change (P5):** In (Zhu et al. (2023)), the authors showed that the adversarial perturbations in the S-channel (or saturation channel) of an image in HSV (Hue Saturation Value) form ensures a high success rate for attacks compared to other channels. In our work, we increase the saturation of the multi-modal input ten-fold based on the intuition that a subtle change can affect the models' predictions.

**d) Composite Perturbation (CP or P6):** We consider a composite case where TSFM is combined with another AI system to reflect the influence of market sentiment on stock prediction (Mishev et al. (2020)). We assess sentiment for each time series by passing the corresponding time-series line plot to a zero-shot CLIP-based sentiment analysis system (SAS) (Radford et al. (2021); Bondielli et al. (2021)), which outputs sentiment intensity values (negative (-1), neutral (0), positive (1)). These labels are scaled to [0, 255] and represented as a sentiment intensity stripe. This zero-shot CLIP-based SAS may exhibit bias or inaccuracies; our goal is to study the robustness of MM-TSFM in conjunction with such an AI system.

### 4.2 Evaluation Metrics

In this section, we describe our evaluation metrics for measuring robustness and forecasting accuracy.

#### 4.2.1 Robustness Metrics

We adapt the Weighted Rejection Score (WRS) originally proposed in (Lakkaraju et al. (2024)) to measure statistical bias. Additionally, we introduce two new metrics: APE and PIE % (modified versions of ATE (Abdia et al. (2017)) and DIE % (Lakkaraju et al. (2024)) tailored to answering our research questions:

**Weighted Rejection Score (WRS):** WRS quantifies statistical bias across protected attributes ($Z$) by assessing the extent to which the null hypothesis is rejected at various confidence intervals (CIs). For a given protected attribute $z_i \in Z$, outcome distributions ($R_t^{max}|z_i$) are compared across all pairs of groups within $z_i$ (for e.g., if $z_i$ has $m$ protected groups, $^mC_2$ pairwise comparisons are performed). The t-value for each pair is computed using Student's t-test (Student (1908)). Let $x_i$ represent the number of rejections for different CI, and $w_i$ be the weight assigned to that CI. Specifically, weights of 1, 0.8, and 0.6 are used for 95 %, 75 %, and 60 % CIs, respectively. WRS is given by:

$$WRS = \sum_i w_i * x_i \tag{1}$$

**Average Perturbation Effect (APE)**: In causal inference, Average Treatment Effect (ATE) provides the average difference in outcomes between between treated and untreated units (Wang et al. (2017)). In our context, it computes the difference between perturbed data residuals (P1 through P6) and the unperturbed data residuals (P0), thereby measuring the impact of the perturbation on the outcome. Hence, we refer to this metric as APE. It is formally defined using the following equation:

$$[|E[R_t^{max} = j|do(P = i)] - E[R_t^{max} = j|do(P = 0)]|] \tag{2}$$

**Propensity Score Matching - Deconfounding Impact Estimation % (PSM-DIE % or PIE %)** In (Lakkaraju et al. (2024)), a linear regression model was used to estimate causal effects, assuming a linear relationship between variables. This method, however, doesn't capture non-linear relationships or fully eliminate confounding biases and only works for binary treatments. Our work uses six treatment (perturbation) values, applying Propensity Score Matching (PSM) (Rosenbaum & Rubin (1983)) to target confounding effects by matching treatment and control units based on treatment probability, similar to RCTs and independent of outcome variables (Baser (2007)). It is defined as:

$$[||APE_o| - |APE_m||] * 100 \tag{3}$$

$APE_o$ and $APE_m$ represent APE computed before and after applying PSM, respectively. $PIE\%$ measures the true impact of $Z$ on the relationship between $P$ and $R_t^{max}$.

### 4.2.2 Forecasting Accuracy Metrics

We evaluate forecasting accuracy using three metrics (Makridakis et al. (2022)):

**Symmetric mean absolute percentage error (SMAPE)** measures the relative difference between predicted and actual values and assigns equal weight to over- and under-estimations. It is defined as,

$$SMAPE = \frac{1}{T} \sum_{t=1}^{T} \frac{|x_t - \hat{x}_t|}{(|x_t| + |\hat{x}_t|)/2}, \tag{4}$$

where $T = 20$ (i.e., the value of $d$) is the total number of observations in the predicted time series. SMAPE scores range from 0 to 2, with lower scores indicating more precise forecasts.

**Mean absolute scaled error (MASE)** measures the mean absolute error of forecasts relative to that of a naive one-step forecast on the training data.

$$MASE = \frac{\frac{1}{T} \sum_{i=t+1}^{t+T} |x_i - \hat{x}_i|}{\frac{1}{t} \sum_{i=1}^{t} |x_i - x_{i-1}|}, \tag{5}$$

where in our case, $t = 80$, and $T = 20$. Lower MASE values indicate better forecasts.

**Sign Accuracy** quantifies the proportion of correctly predicted directional changes in the time series. A higher accuracy indicates better alignment with actual trend movements.

$$\text{Sign Accuracy} = \frac{1}{T} \sum_{t=1}^{T} 1\left(\text{sign}(\hat{x}_t - \hat{x}_{t-1}) = \text{sign}(x_t - x_{t-1})\right), \tag{6}$$

where $T$ is the total number of predicted time steps, $x_t$ is the actual value, and $\hat{x}_t$ is the predicted value at time $t$. The indicator function $1(\cdot)$ returns 1 if the predicted sign matches the actual sign and 0 otherwise. Higher Sign Accuracy values indicate better directional prediction.

### 4.3 Workflow

Our proposed workflow consists of two components: *Data to Predictions* and *Predictions to Ratings*. In the first stage, as shown in Figure 3a, TSFM processes the input and predicts the next 'd' timesteps. The TSFM and baseline models are detailed in Section 5.1.1. In the second stage, illustrated in Figure 3b, we extend the approach from (Lakkaraju et al. (2024)), originally designed for assessing SASs, to accommodate our more complex multi-modal data with multiple perturbations, beyond the original textual data and binary

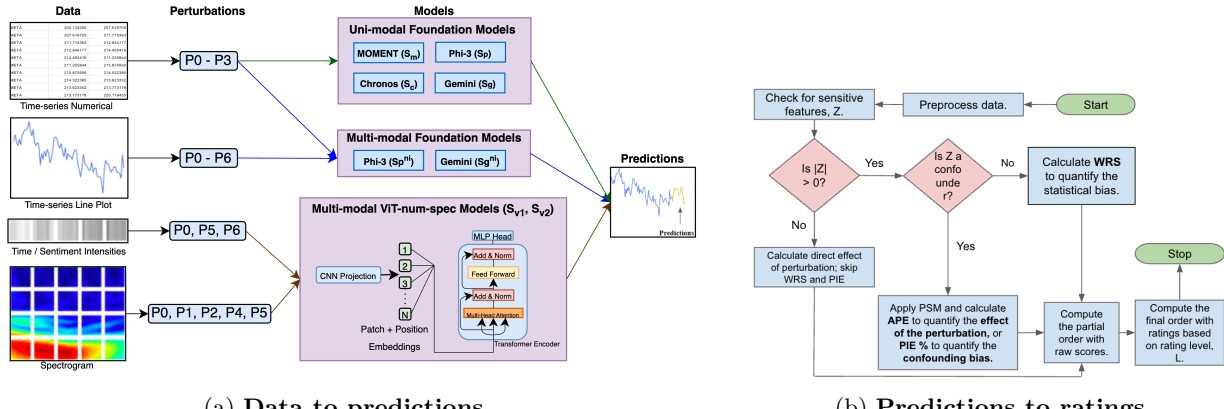

(a) **Data to predictions**          (b) **Predictions to ratings**

Figure 3: (a) Workflow illustrating the data modalities, the perturbations applied to each, and their propagation through different models to generate predictions. P0 is unperturbed, P1 (Drop-to-zero) and P2 (Value halved) are semantic perturbations (SP), P3 (Missing Values) is a syntactic perturbation (SyP), P4 (Single pixel change) and P5 (Saturation change) are input-specific perturbations (IP), and P6 is a composite perturbation (CP). (b) Workflow for performing statistical and causal analysis to compute raw scores and assign final ratings to the test systems.

treatments. The modified metrics, APE and PIE %, introduced in Section 4.2, help manage this complexity. These raw scores establish a partial order for determining final model ratings, which vary based on the rating level, $L$. The following five algorithms provide the detailed implementation of metrics, raw score calculation, and the final ratings calculation (Algorithms 1, 2, and 3 are provided in the Appendix A):

- **Algorithm 1** computes WRS, as defined in Section 4.2, to measure statistical bias by analyzing how sensitive the model's outcomes are to variations in sensitive attributes. **This helps us answer RQ1 from Section 3.**

- **Algorithm 2** calculates the PIE %, as defined in Section 4.2, to assess confounding bias by measuring the effect of the confounder on model outcomes before and after deconfounding. **This can help us answer RQ2 from Section 3.**

- **Algorithm 3** calculates APE by evaluating the difference in the model's outcomes for perturbed data and unperturbed data to evaluate the impact of the perturbation on the outcome as defined in Section 4.2. **This helps us answer RQ3 from Section 3.**

- **Algorithm 4** generates a partial order of TSFM for each perturbation based on their WRS or PIE or ATE scores. Models are ranked based on these raw scores, and the rankings are stored in a dictionary mapping each perturbation to its corresponding system order (as shown in Tables 3 and 4).

- **Algorithm 5** assigns final ratings to systems for each treatment based on the partial order generated by Algorithm 4. It partitions the raw scores within each treatment into $L$ user-defined rating levels and assigns ratings accordingly. The output is a dictionary where treatments serve as keys, mapping to model ratings as values. Lower ratings indicate better performance and greater robustness, except for Sign Accuracy, where higher ratings correspond to better performance.

## 5   Experiments and Results

This section introduces the TSFM used in our experiments, baseline models, test data, and evaluation metrics, including two new metrics for perturbations and confounders. We also present the user study design, responses, and findings.

---

**Algorithm 4:** *CreatePartialOrder*

---

**Purpose:** Create a partial order of systems within each treatment based on raw scores (APE/ATE, PIE, or WRS).
**Input:**
$S$, $d$ (as defined in the previous algorithm); $P$, Set of treatments; $Metric$, specifies which metric to use (APE/ATE, PIE, or WRS).
**Output:**
$PO$, dictionary with partial orders for each treatment.
$PO \leftarrow \{\}$          // Initialize dictionary for partial orders.
$SD \leftarrow \{\}$
**for** *each $p_i \in P$* **do**
     $SD \leftarrow \{\}$          // Reset scores for each treatment.
     **for** *each $s_j \in S$* **do**
         **if** $Metric == APE/ATE$ **then**
            $\psi \leftarrow ComputeATEScore(s_j, d, p_i, p_0)$          // Compute ATE.
         **else**
            **if** $Metric == PIE$ **then**
                $\psi \leftarrow ComputePIEScore(s_j, d, p_i, p_0)$          // Compute PIE.
            **else**
                $\psi \leftarrow WeightedRejectionScore(p_i, s_j, d)$          // Compute WRS.
            **end**
         **end**
         $SD[s_j] \leftarrow \psi$          // Store score for system $s_j$.
     **end**
     $PO[p_i] \leftarrow SORT(SD)$          // Sort systems in ascending order of scores.
**end**
**return** $PO$

---

**Algorithm 5:** *AssignRating*

---

**Purpose:** Assign a rating to each system based on the partial order and the number of rating levels, $L$.
**Input:**
$S$, $d$, $P$, $Metric$ (as defined in the previous algorithm); $L$, rating levels chosen by the user.
**Output:**
$R$, dictionary with perturbations as keys and ratings for each system as values.
$R \leftarrow \{\}$;
$PO \leftarrow$ CreatePartialOrder($S, d, P, Metric$);
**for** $p_i \in P$ **do**
     $\psi \leftarrow [PO[p_i].values()]$;
     **if** $len(S) > 1$ **then**
         $G \leftarrow$ ArraySplit($\psi, L$);
         $SD \leftarrow \{\}$;
         **for** $k, i \in PO[p_i]$ **do**
            $SD[k] \leftarrow$ FindGroup($i, G$);
         **end**
     **end**
     **else**
         // Case of a single SAS in $S$
         **if** $\psi == 0$ **then**
            $SD[k] \leftarrow 1$;
         **end**
         **else**
            $SD[k] \leftarrow L$;
         **end**
     **end**
     $R[p_i] \leftarrow SD$;
**end**
**return** $R$;
**FindGroup Function:**
**Input:** $value$, $groups$; **Output:** Group index.
**for** $g_j \in groups$ **do**
     **if** $value \in g_j$ **then**
         **return** index of $g_j$;
     **end**
**end**

---

### 5.1 Experimental Apparatus

### 5.1.1 Test Models

We evaluate six foundation models (FMs) and three baseline models. Among the FMs, four are used directly, while two (Gemini-V and Phi-3) have both uni-modal and multi-modal variants, making them distinct models. Table 1 provides an overview of the FM architectures. We also consider two ViT-num-spec models, which are vision transformer-based models trained for time-series forecasting task.

| Model | Mode | Size | Purpose & Arch. | Inf. Time (sec/sample) |
|---|---|---|---|---|
| Gemini 1.5 Flash | Multi | 32B* | GP-1A, 1B, Decoder | 1.6 (1A); 10.2 (1B) |
| Phi-3-vision | Multi | 4.2B | GP-2A, 2B, Enc-Dec | 19.7 (1A); 26.6 (1B) |
| MOMENT-large | Uni | 385M | TS-1, Encoder | 0.315 |
| Chronos-T5-small | Uni | 46M | TS-2, Enc-Dec | 0.811 |
| ViT-num-spec | Multi | 86M | TS-3, Encoder | 6 |

Table 1: Overview of the architectural details of TSFM. *Best guess in the absence of official information.

**Foundation Models:**

**1. MOMENT** ($S_m$) (Goswami et al. (2024)) is an open-source FM for forecasting, classification, anomaly detection, and imputation in zero-shot and few-shot settings. It is based on the T5-Large encoder (Raffel et al. (2020)) and can be fine-tuned if needed.

**2. Chronos** ($S_c$) (Ansari et al. (2024)) is a pretrained probabilistic time-series model that tokenizes time-series values using scaling and quantization. It employs a T5 encoder-decoder and is trained via cross-entropy loss. We use Chronos-T5-Small.

**3. Gemini-V** ($S_g$, $S_g^{ni}$) (Team et al. (2023)) is a multi-modal FM designed to process both text and images. We use $S_g$ (numeric-only mode) that processes only numerical time-series data, and $S_g^{ni}$ (numeric + vision mode) that processes numerical data and time-series line plots.

**4. Phi-3** ($S_p$, $S_p^{ni}$) (Abdin et al. (2024)) is a lightweight, state-of-the-art multi-modal FM. We use $S_p$ (numeric-only mode) that processes only numerical time-series data, and $S_p^{ni}$ (numeric + vision mode) that processes numerical data and time-series line plots. Below is the prompt template used for time-series forecasting with Gemini-V and Phi-3 models (text highlighted in red is omitted for uni-modal forecasting):

**Prompt to Uni-modal and Multi-modal FMs**

*"You are a time series forecasting model that only outputs the forecasted numerical values." "**Input: <time series>**" "Given the input time series for the past 80 time steps and the corresponding time series plot, can you forecast the next 20 time steps? Provide a list of 20 numeric values only. Do not provide any discussion."*

**Prompt to Multi-modal FMs for P6**

*"You are a time series forecasting model that only outputs the forecasted numerical values." "**Input: <time series>**" "Given the input time series from the past 80 time steps and the corresponding sentiment intensity plot, where darker shades indicate more negative sentiment and lighter shades indicate more positive sentiment, Provide a list of 20 numeric values only. Do not provide any discussion."*

**5. ViT-num-spec Models** ($S_{v1}$, $S_{v2}$): We employ the **ViT-num-spec** model (Zeng et al. (2023)), which combines a **vi**sion **t**ransformer with a multimodal time-frequency **spec**trogram, augmented by the intensities of **num**eric time series for time series forecasting. This model improves predictive accuracy by leveraging both visual and numerical data. Specifically, it transforms numeric time series into images using a time-frequency spectrogram and utilizes a vision transformer (ViT) encoder with a multilayer perceptron (MLP) head for future predictions.

**Time-Frequency Spectrogram**: Building on the method of (Zeng et al. (2023)), we use wavelet transforms (Daubechies (1990)) to create time-frequency spectrograms from time series data. Specifically, we employ the Morlet wavelet (Scipy (2024)) with scale $s$ and central frequency $\omega_0 = 5$, as detailed in Equation 7:

$$\psi(x) = \sqrt{\frac{1}{s}} \pi^{-\frac{1}{4}} \exp\left(-\frac{x^2}{2s^2}\right) \exp\left(j\omega_0 \frac{x}{s}\right) \tag{7}$$

This method convolves the time series with wavelets at various scales, producing coefficients that indicate signal strength at different frequencies. These magnitudes are visualized in a spectrogram, with higher frequencies at the top and lower ones at the bottom. To retain sign information, a stripe from the standardized

numeric time series is added to the top of the spectrogram image. This enhanced image is then used as input for the vision transformer model.

**Vision Transformer**: In the next stage, the ViT-num-spec uses a vision transformer with an MLP head for time series forecasting. Input images are segmented into 16 x 16 non-overlapping patches, projected into tokens, and augmented with 1D positional embeddings. The encoder converts these patches into latent representations. For our implementation, 128 x 128 images have price movements in a 16 x 128 top row, with the spectrogram occupying the remaining 112 x 128 space below.

We trained two variations of the ViT-num-spec model using two real-world datasets and conducted evaluations using a separate dataset. $S_{v1}$ (Pre-COVID training) was trained on S&P 500 stock data from 2000-2014 (46,875 training samples, 46,857 validation samples). $S_{v2}$ (COVID-period training) was trained on data from March 2020–November 2022 (7,478 training samples, 7,475 validation samples).

### 5.1.2 Baselines

We consider the following baselines:

**1. Auto Regressive Integrated Moving Average (ARIMA)** ($S_a$) is a widely used statistical approach for time series forecasting. It combines three different components: Autoregressive (AR), differencing (I), and moving average (MA) to capture the patterns in the time-series data and predict the next 'd' (from section 3) values.

**2. Biased system** ($S_b$) is an extreme baseline biased towards META and GOOG (technology companies), assigning residuals of 0 and 200 respectively, while assigning higher residuals to other companies, representing maximum bias.

**3. Random system** ($S_r$) assigns random price predictions within a company range for contextually meaningful values.

### 5.1.3 Test Data

We collected daily stock prices from Yahoo! Finance for six companies across different industries: Meta (META) and Google (GOO) in social technology, Pfizer (PFE) and Merck (MRK) in pharmaceuticals, and Wells Fargo (WFC) and Citigroup (C) in financial services. The data spans from March 28, 2023, to April 22, 2024. We used data from March 28, 2023, to March 22, 2024, to predict stock prices for the following month.

## 5.2 Experimental Evaluation

In this section, we describe the experimental setup used to address the RQs stated in Section 3, the results obtained, and the conclusions drawn from the results. Figure 1b shows the causal diagrams used to answer the RQs.

**RQ1:** **Does *Sensitive Attribute* affect the *Residual*, even though *Sensitive Attribute* has no effect on *Perturbation*?**

**Setup:** In this experiment, the causal link from the *Sensitive Attribute* to *Perturbation* is absent, as the perturbation to the stock prices does not depend on the corresponding company name or the industry, i.e., perturbations are applied uniformly across all the data points. We quantify the statistical bias exhibited by the systems by using WRS described in Section 4.2. We perform two different analyses in this experiment: one to measure the discrepancy shown across various industries ($\text{WRS}_{Industry}$) and another to measure the discrepancy among both the companies ($\text{WRS}_{Company}$) within the same industry.

**Results and conclusion:** From Table 2 (and Figure 6 in the Appendix), most discrepancies can be observed across industries (inter-industry) compared to the discrepancies across companies within each industry. When the input data was subjected to perturbation P2 ($WRS_{avg}$ of 5.55), the systems exhibited more statistical bias. From Figure 4 and Table 2, $S_a$ ($WRS_{avg}$ of 3.96) exhibited the least statistical bias, while $S_p$ ($WRS_{avg}$ of 6.27) exhibited the highest statistical bias among the systems evaluated under the perturbations

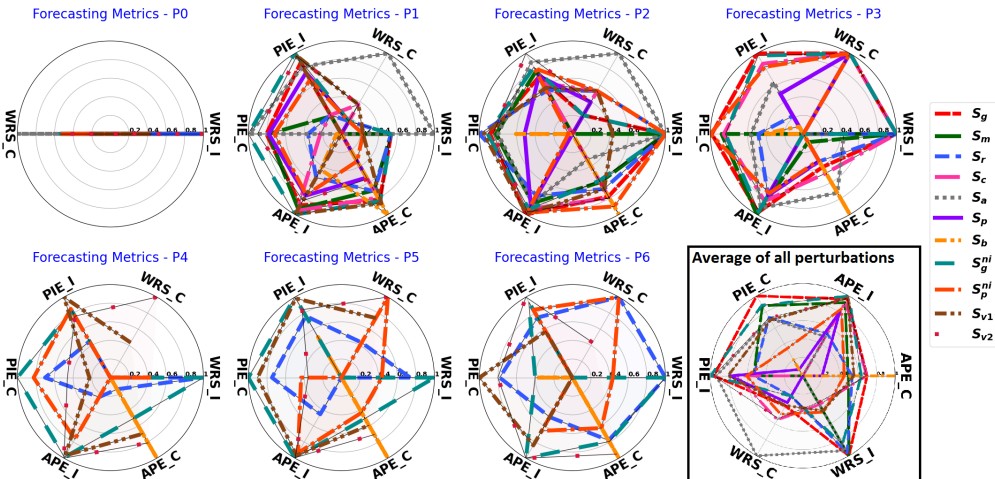

Figure 4: Radar plots showing robustness metrics for all models under different perturbations (P1 through P6) and their average (bottom right). **Lower values (i.e., lines closer to the center) indicate lower robustness, while points farther from the center represent better robustness across metrics.**

considered. Hence, we conclude that *Sensitive Attribute* affects the *Residual*, even though *Sensitive Attribute* has no effect on *Perturbation*.

**RQ2: Does *Confounder* affect the relationship between *Perturbation* and *Residual*, when *Confounder* has an effect on *Perturbation*?**

**Setup:** In this experiment, we use PIE % defined in equation 3 to compare the APE (defined in equation 2) before and after deconfounding using the PSM technique as the presence of the confounder opens a backdoor path from *Perturbation* to *Residual* through the *Confounder*. The causal link from *Confounder* to *Perturbation* will be valid only if the perturbation applied depends on the value of the confounder (i.e. the company or the industry the specific data points belong to). To ensure the probability of perturbation assignment varies with respect to the *Confounder* across three distributions (DI1 through DI3) in the case of *Industry* and six different distributions in the case of *Company* (DC1 through DC6), we implement weighted sampling. For each distribution, weights are configured so that perturbation groups P1 through P6 have a twofold higher likelihood of selection compared to P0 for specific values of the confounder. For example, META in DC1, GOOG in DC2, and so on. This strategy highlights significant cases, although other combinations are possible for further exploration.

**Results and Conclusion:** Figure 6 in the Appendix shows that selecting *Industry* as the confounder leads to greater confounding bias in the systems. In Figure 4, $S_g^{ni}$ ($PIE_{avg}\%$ of 1107.66) exhibited the least confounding bias, while $S_p^{ni}$ ($PIE_{avg}\%$ of 2778.06) exhibited the most. Systems showed more confounding bias under perturbation P5 ($PIE_{avg}\%$ of 3646.20). Therefore, the *Confounder* affects the relationship between *Perturbation* and *Residual*, particularly when the *Confounder* influences the *Perturbation*.

**RQ3: Does *Perturbation* affect the *Residual* when *Sensitive Attribute* may have an effect on *Residual*?**

**Setup:** The experimental setup in this experiment is the same as that for answering RQ2. To compute the APE, we used PSM described in Section 4.2. PSM allows us to effectively determine the effect of *Perturbation* on the *Residual*. For instance, if two matched points belong to the same company but only one was perturbed, any difference in their residuals can be directly attributed to the perturbation itself rather than to other confounding factors. This method provides a clear understanding of the true impact of the *Perturbation* on the *Residual*. As our rating method aims to bring out the worst possible behavior of the systems, we take the MAX(APE) as the raw score that is used to compute the final ratings.

**Results and Conclusion:** It is undesirable to have a higher APE, as it implies that the perturbation applied can have a significant impact on the residuals of different systems. From Figure 6 in the Appendix, when *Industry* was considered as the confounder, it led to a higher APE. As the outcome of $S_b$ depended on the *Company* (and varied from one company to another), the perturbation did not have any effect on the system. Whereas, when *Industry* was considered as the confounder, the perturbation appeared influential, resulting in a high APE for $S_b$. From Figure 4 and Table 2, perturbations had the least impact on $S_g^{ni}$ ($APE_{avg}$ of 5.89) and highest impact on $S_a$ ($APE_{avg}$ of 27.73). Among all the perturbations, P1 ($APE_{avg}$ of 20.25) was the most disruptive. Hence, *Perturbation* affects the *Residual* when *Sensitive Attribute* may have an effect on *Residual*.

**RQ4:** Does *Perturbations* degrade the accuracy of $S$?

**Setup:** In this experiment, we compute the three accuracy metrics widely used in for the task of financial time-series forecasting (Makridakis et al. (2022)), which were summarized in Section 4.2.

**Results and Conclusion:** Radar plots for the accuracy metrics are shown in Figure 7 in the Appendix. Overall, $S_c$ exhibited the highest amount of forecasting accuracy in terms of SMAPE (average of 0.05) and MASE (average of 4.67), while $S_a$ outperformed all other systems in terms of sign accuracy (average of 58.57). $S_b$ consistently predicted the correct directional movement of stock prices, exhibiting high sign accuracy as it was designed to adjust residuals based on specific company stock prices. Perturbation P6 caused the highest decline in SMAPE (average of 0.39) and MASE (average of 176.43), while P1 caused the highest decline in sign accuracy (average of 49.86). Hence, *Perturbations* degrade the accuracy of *S*.

## 5.3 Overall Performance Comparison

Now, we provide an overall comparison of the different systems across all metrics to highlight key findings about their performance under various perturbations by answering the additional research questions (ARQs) stated in Section 1. Figures 4 and 7 (in the Appendix) show radar plots with robustness metrics and forecasting accuracy, respectively.

**Clear Domination Signals:** From Figure 4 and the detailed results from Section 5.2, we can draw the following conclusions:

$S_c$**'s Superiority in Forecasting Metrics**: $S_c$ consistently outperformed other models in terms of forecasting accuracy metrics, specifically SMAPE and MASE. This indicates that $S_c$ is highly effective in predicting stock prices with minimal error.

**General Superiority Over Biased and Random Systems**: All models perform better than the biased and random systems in forecasting metrics. This underscores the importance of using well-designed models over naive or biased approaches.

**Robustness in PIE % and APE Metrics**: According to the average scores, the $S_g^{ni}$ system demonstrated superior robustness in PIE % and APE metrics. This suggests that $S_g^{ni}$ is more resilient to perturbations and confounding biases compared to other systems.

**Role of Confounders**: Our analysis (Figs. 6 and 4 in the Appendix) shows that using industry as a confounder introduces more bias, with higher PIE% and APE scores indicating significant industry-specific effects on the relationship between perturbations and residuals. Inter-industry comparisons also show more discrepancies, as evidenced by WRS scores.

**ARQ1: How do different model types compare in performance? (Foundation vs. Non-foundation)**

**Answer:** As shown in Figure 5a, Non-foundation models (ViT-num-spec models) demonstrate superior performance over foundation models across both robustness and forecasting accuracy metrics.

**ARQ2: Does multi-modality improve the performance of TSFM?**

**Answer:** The results in Figure 5b show the effect of multi-modality on model performance. While $S_g$ (uni-modal) outperforms $S_g^{ni}$ (multi-modal) and $S_p^{ni}$ (multi-modal) outperforms $S_p$ (uni-modal) in terms of

| Research Question | Causal Diagram | Metrics Used | Comparison across Systems | Comparison across Perturbations | Key Conclusions |
|---|---|---|---|---|---|
| **RQ1:** Does $Z$ affect $R_t^{max}$, even though $Z$ has no effect on $P$? |  | WRS | {$S_a$: 3.96, $S_g$: 5.05, $S_r$: 5.15, $S_{v2}$: 5.20, $S_g^{ni}$: 5.44, $S_c$: 5.46, $S_p^{ni}$: 5.48, $S_{v1}$: 5.71, $S_m$: 5.75, $S_p$: 6.27, $S_b$: 6.9} | {P4: 5.42, P1: 5.49, P3: 5.51, P5: 5.52, P6: 5.52, P2: 5.55, P0: 5.70} | **$S$ with low statistical bias:** $S_a$. **$S$ with high statistical bias:** $S_p$. **$P$ that led to more statistical bias:** P0 **Analysis with more discrepancy:** Inter-industry |
| **RQ2:** Does $Z$ affect the relationship between $P$ and $R_t^{max}$ when $Z$ has an effect on $P$? |  | PIE % | {$S_g^{ni}$: 1107.66, $S_g$: 1115.08, $S_{v2}$: 1346.46, $S_a$: 1448.29, $S_{v1}$: 1848.20, $S_p$: 2459.30, $S_m$: 2544.20, $S_r$: 2668.52, $S_c$: 2755.50, $S_p^{ni}$: 2778.06, $S_b$: 4758.16} | {P1: 1711.88, P4: 2035.95, P3: 2057.31, P6: 2410.28, P2: 2628.52, P5: 3646.20} | **$S$ with low confounding bias:** $S_g^{ni}$. **$S$ with high confounding bias:** $S_p^{ni}$. **$P$ that led to more confounding bias:** P5. **Confounder that led to more bias:** *Industry* |
| **RQ3:** Does $P$ affect $R_t^{max}$ when $Z$ may have an effect on $R_t^{max}$? |  | APE | {$S_g^{ni}$: 5.89, $S_{v1}$: 7.34, $S_c$: 7.80, $S_m$: 9.83, $S_g$: 6.46, $S_{v1}$: 6.46, $S_p^{ni}$: 12.66, $S_p$: 15.98, $S_r$: 21.57, $S_a$: 27.73, $S_b$: 33.95} | {P3: 9.96, P6: 12.2, P2: 12.9, P4: 13.19, P5: 18.36, P1: 20.25} | **$S$ with low APE:** $S_g^{ni}$. **$S$ with high APE:** $S_a$. **$P$ with low APE:** P3. **$P$ with high APE:** P1. **Confounder that led to high APE:** *Industry* |
| **RQ4:** Does $P$ affect the accuracy of $S$? | This hypothesis does not necessitate a causal model for its evaluation. | SMAPE, MASE, Sign Accuracy | **SMAPE:** {$S_c$: 0.051, $S_{v1}$: 0.053, $S_g$: 0.055, $S_a$: 0.058, $S_{v2}$: 0.06, $S_g^{ni}$: 0.06, $S_r$: 0.83, $S_p^{ni}$: 0.084, $S_p$: 0.097, $S_m$: 0.098, $S_b$: 1.276}; **MASE:** {$S_c$: 4.67, $S_{v1}$: 4.80, $S_g$: 5.04, $S_{v2}$: 5.49, $S_g^{ni}$: 5.76, $S_a$: 8.54, $S_p^{ni}$: 7.60, $S_p$: 9.02, $S_m$: 9.13, $S_r$: 86.76, $S_b$: 947.56 }; **Sign Accuracy:** {$S_m$: 40.91, $S_p$: 44.42, $S_p^{ni}$: 45.24, $S_{v2}$: 49.33, $S_r$: 49.75, $S_g$: 50.93, $S_{v1}$: 51.34, $S_g^{ni}$: 51.37, $S_c$: 51.99, $S_a$: 58.57, $S_b$: 62.6} | **SMAPE:** {P0: 0.24, P2: 0.25, P1: 0.26, P3: 0.28, P4: 0.38, P5: 0.38, P6: 0.39}; **MASE:** {P0: 99.06, P2: 99.57, P1: 101.27, P3: 119.66, P4: 175.56, P5: 175.56, P6: 176.43}; **Sign Accuracy:** {P1: 49.86, P0: 51.35, P2: 51, P3: 51.16, P4: 51.79, P5: 51.43, P6: 50}; | **$S$ with good performance:** $S_c$. **$S$ with poor performance:** $S_m$. **$P$ with high impact on performance:** P6. |

Table 2: Summary of the research questions answered in the paper, causal diagram, metrics used in the experiment, average of the metric values compared across different systems, average computed across different perturbations, and the key conclusions drawn from the experiment. All the raw scores and ratings are shown in Tables 3 and 4.

individual metrics, the overall average trend across all metrics suggests that incorporating multiple modalities can lead to more balanced and improved performance in both robustness and accuracy.

**ARQ3: How does the architecture of FMs influence performance? (Time-series-specific vs. general purpose and encoder-only vs. decoder-only vs. encoder-decoder)**

**Answer:** Our evaluation (Figure 5c, left) indicates that the Time Series architecture generally performs better across several metrics, such as achieving the best values in APE_C, PIE_C, SMAPE, MASE, and WRS_I, suggesting that the TS architecture may be more effective for these specific tasks compared to the general purpose architectures. Figure 5c (right) shows that decoder-only architecture outperforms others in terms of both accuracy and robustness.

Overall, our rating method highlights $S_c$'s forecasting accuracy, the robustness of multimodal systems against perturbations and confounding biases, and the superiority of well-designed models over naive approaches.

## 5.4 User Study

We conducted a user study to evaluate the ratings generated by our approach for comparing the behavior of various TSFM based on two key metrics: robustness and statistical fairness (defined as lack of statistical bias). To simplify the evaluation for participants, we converted the generated ratings into rankings (i.e., the system with the highest robustness ranking is the most robust system). The main objective of this study was to validate the following hypotheses:

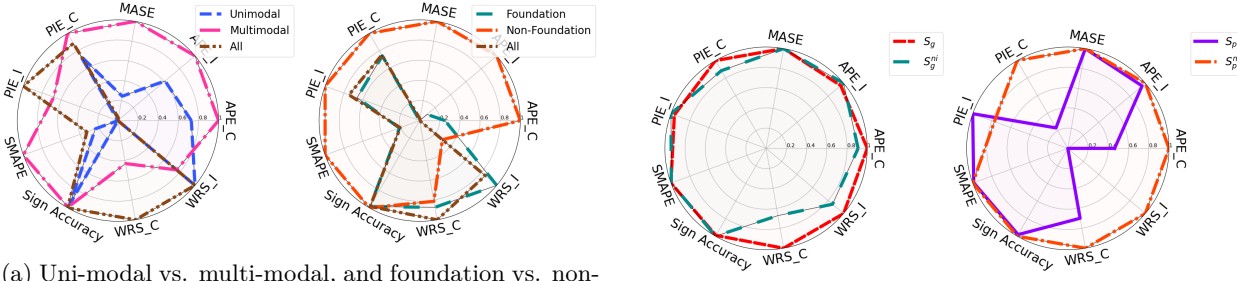

(a) Uni-modal vs. multi-modal, and foundation vs. non-foundation models.

(b) Effect of modalities for $S_g$ (left) and $S_p$ (right).

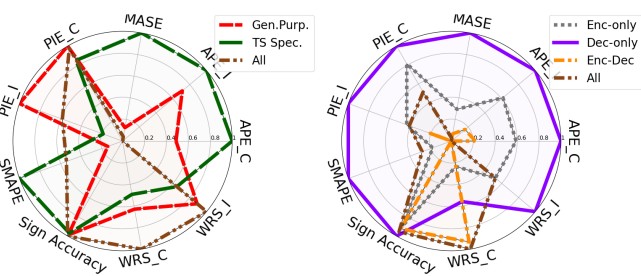

(c) Role of architecture in forecasting accuracy and robustness.

Figure 5: Radar plots comparing model performance across different dimensions. (a) compares uni-modal vs. multi-modal and foundation vs. non-foundation models, (b) shows the effect of modality for selected systems, and (c) analyzes the role of architecture. **Lower values (i.e., lines closer to the center) indicate lower performance, while points farther from the center represent better performance across metrics.** See Table 1 for model details.

**HP1:** Rankings generated by our approach decrease the difficulty of comparing system robustness.

**HP2:** Rankings generated by our approach decrease the difficulty of comparing system fairness (lack of statistical bias).

**HP3:** Rankings generated by our method align with users' rankings for both fairness and robustness.

This IRB-approved study[1] involved participants being presented with TSFM models predicting the future stock prices for companies across different industries, such as Technology and Pharmaceuticals. To ensure informed evaluations, we introduced participants to key concepts including robustness, fairness, and error metrics (maximum residual, mean, and standard deviation of errors) regardless of prior knowledge.

The study was structured into four panels: a *self-assessment* on knowledge about time series and financial data, a *fairness* panel, where fairness was the evaluated metric, and two *robustness* panels, where robustness was assessed under two different perturbations (P1 and P2). To make the plots and rankings easier to interpret for participants, we selected two perturbations (P1 and P2) from the six used in our experiments and six representative systems from the original set of eleven.

In the fairness panel, participants were presented with graphs depicting the residual values of six different systems and an ideal system using stock price data from the Technology and Pharmaceuticals industries. They were provided with the mean and standard deviation of errors and asked to rank the systems from least to most fair. Participants then rated the difficulty of this ranking task (1 being the most difficult). Subsequently, they were shown the rankings generated by our approach and asked to rate the accuracy of these rankings (1 being the least accurate). Finally, participants were asked to rank the difficulty of comparing the behavior of different systems using our rankings (1 being the most difficult).

---

[1]Details anonymized for reviewing.

For the robustness panels, similar questions were posed, with users evaluating systems based on their robustness to different perturbations (P1 and P2). A total of 26 users from academia and industry participated over two weeks. We performed different types of statistical tests (results are shown in Tables 5 and 6 in the Appendix) to draw conclusions. We now discuss the key findings from the tests.

To evaluate *HP1*, we conducted a paired t-test to compare user responses on difficulty of ranking various systems before and after presenting our rankings. The same participants assessed the difficulty using both the graph representation of fairness and our ranking representation, making the paired t-test appropriate. Paired t-test also accounts for the inherent correlation between the paired rankings, making it suitable to account for potential different perceptions across the two representations. Paired t-tests for each robustness panel indicated a significant difference before (P1:$\mu = 2.70$, $\sigma = 1.06$; P2: $\mu = 2.65$, $\sigma = 1.17$) and after (P1: $\mu = 3.23$, $\sigma = 1.42$; P2: $\mu = 3.07$, $\sigma = 1.44$) our rankings were presented with P1:$t(26) = -1.89$, $p = 0.030$, and P2: $t(26) = -1.62$, $p = 0.059$. Since the p-values $\prec 0.1$, we confirm *HP1* that the ranking generated by our approach significantly reduced the perceived difficulty of comparing different systems. Same approach is used to evaluate *HP2*. The paired t-test showed no significant change in perceived difficulty scores before ($\mu = 2.54$, $\sigma = 1.30$) and after ($\mu = 2.92$, $\sigma = 1.35$) our rankings were presented, $t(26) = -1.18$, $p = 0.12$. Since $p \succ 0.1$, we conclude that our ranking representation did not significantly reduce the perceived difficulty of comparing different systems. The lack of significant reduction in perceived difficult may have stemmed from the complexity of the graphical representations. To validate *HP3*, we used the Spearman Rank Correlation coefficient (Zar (2005)) ($\rho$) to evaluate the alignment between the users' rankings and those produced by our approach. We considered a confidence interval of 90 %. The fairness panel showed a high correlation ($\rho = 0.73$), and the robustness under P1 showed a strong correlation ($\rho = 0.91$). However, robustness under P2 showed a weak correlation ($\rho = 0.14$).

In summary, the results of the user study indicate that the rankings generated by our approach can significantly reduce the difficulty of comparing the robustness of different TSFM systems. However, when the comparison metric is fairness, this reduction is not significant. Additionally, while user rankings align well with our method generated rankings for fairness and robustness under P1, they show a weak correlation for robustness under P2, indicating differing perceptions of the 'value halved' perturbation. P1 (drop-to-zero) involves a significant semantic change that is easier to spot, whereas P2 (value halved) is also a semantic perturbation but subtler, making it potentially harder to identify. Our current study is preliminary and promising; an avenue for future work is to conduct it at a larger scale.

## 6 Discussion and Conclusion

Our paper aimed to measure the impact of perturbations and confounders on the outcome of TSFM using stock prices across leading companies and industries. We studied *Industry* and *Company* as confounders, motivated by the intuition that stakeholders rely on learning-based systems for stock purchase decisions and would be interested in knowing if model errors depend on stock price ranges. For example, does a model commit more errors predicting META's stock prices compared to MRK's? To minimize volatility effects, we performed both intra-industry and inter-industry analyses. In future, we plan to study confounders like seasonal trends and financial news. As demonstrated, we believe metrics should be selected based on the questions one wants to answer, rather than relying solely on statistical accuracy. The hypothesis testing approach from (Lakkaraju et al. (2024)), adapted for our work, helped quantify biases and perturbation impacts on test systems. The perturbations used in our analysis have real-world impacts, applicable to both numeric and multi-modal data. While methods like differential evaluation can find the most impactful perturbation variations, we focused on assessing whether simple, subtle perturbations affect TSFM.

**Conclusion** We proposed a causally grounded empirical framework to study TSFM robustness against six input perturbations, evaluating seven state-of-the-art TSFM across six prominent stocks in three industries. Our framework's ratings accurately assessed TSFM robustness and provided actionable insights for model selection and deployment. Experiments showed multi-modal TSFM exhibited greater robustness, while uni-modal TSFM had higher forecasting accuracy. TSFM trained on time series tasks showed better robustness and accuracy compared to general-purpose TSFM. A user study confirmed our ratings effectively convey TSFM robustness to end-users, demonstrating the framework's real-world applicability.

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

## A  Rating Algorithms

Algorithms 1, 3, and 2 describe the computation of WRS, ATE, and PIE used in our experiments.

## B  Experimental Results

Figure 6 shows the impact of company and industry as confounders on the robustness metrics. Figure 7 presents radar plots of accuracy metrics under different perturbation settings. Tables 3 and 4 report the raw scores and ratings based on robustness and accuracy metrics, respectively.

## C  User Study Results

Table 5 summarizes the results of a one-sample right-tailed t-test for the user study, with a hypothesized mean of 2 and a sample size of 26. Table 6 presents the results for the evaluated hypotheses.

---

**Algorithm 1:** *WeightedRejectionScore*

---

**Purpose:** is used to calculate the weighted sum of the number of rejections of null-hypothesis for Dataset $d_j$ pertaining to a system $s$, Confidence Intervals (CI) $ci_k$ and Weights $w_k$.

**Input:**

$d$, dataset corresponding to a specific perturbation; $CI$, confidence intervals (95%, 70%, 60%); $s$, a model belonging to the set of test models, $S$; $W$, weights corresponding to different CIs (1, 0.8, 0.6).

**Output:**

$Z$, Sensitive attribute; $\psi$, weighted rejection score.

$\psi \leftarrow 0$

**for** *each $ci_i, w_i \in CI, W$* **do**

    // $z_a, z_b$ are classes of $Z$

    **for** *each $z_a, z_b \in Z$* **do**

        $t, pval, dof \leftarrow T - Test(z_a, z_b)$;

        $t_{crit} \leftarrow LookUp(ci_i, dof)$;

        **if** $t_{crit} > t$ **then**

            $\psi \leftarrow \psi + 0$;

        **else**

            $\psi \leftarrow \psi + w_i$

        **end**

    **end**

**end**

**return** $\psi$

---

**Algorithm 2:** *ComputePIEScore*

---

**Purpose:** Calculate the Deconfounding Impact Estimation (PIE) using Propensity Score Matching (PSM).

**Input:**

$s$, $d$ (as defined in the previous algorithm); $p$, a perturbation other than $p_0$ (control or no perturbation).

**Output:**

$\psi$, PIE score.

$APE\_o \leftarrow E(R_t^{max}|P = p) - E(R_t^{max}|P = p_0)$     // Observational APE.

$APE\_m \leftarrow E(R_t^{max}|do(P = p)) - E(R_t^{max}|do(P = p_0))$     // Deconfounded APE.

$\psi \leftarrow (APE\_m - APE\_o) * 100$     // Compute PIE.

**return** $\psi$

---

**Algorithm 3:** *ComputeATEScore*

---

**Purpose:** Calculate the Average Treatment Effect (ATE).

**Input:**

$s$, $d$, $p$, $p_0$ (as defined in the previous algorithm)

**Output:**

$\psi$, ATE score.

$\psi \leftarrow E(R_t^{max}|do(P = p)) - E(R_t^{max}|do(P = p_0))$     // Compute ATE.

**return** $\psi$

---

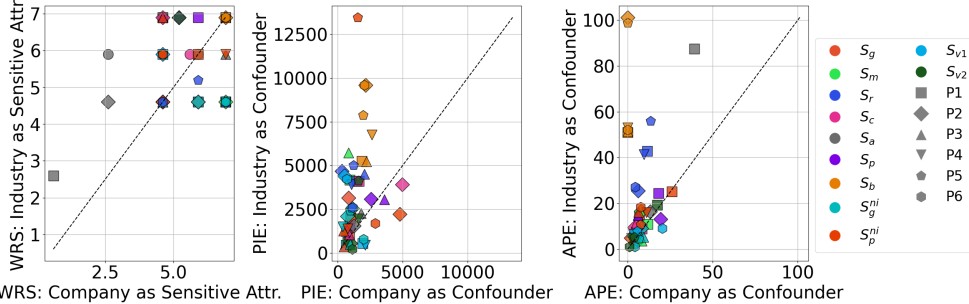

Figure 6: Plots showing the impact of company and industry as confounders for all the robustness metrics considered. Lower values indicate better robustness.

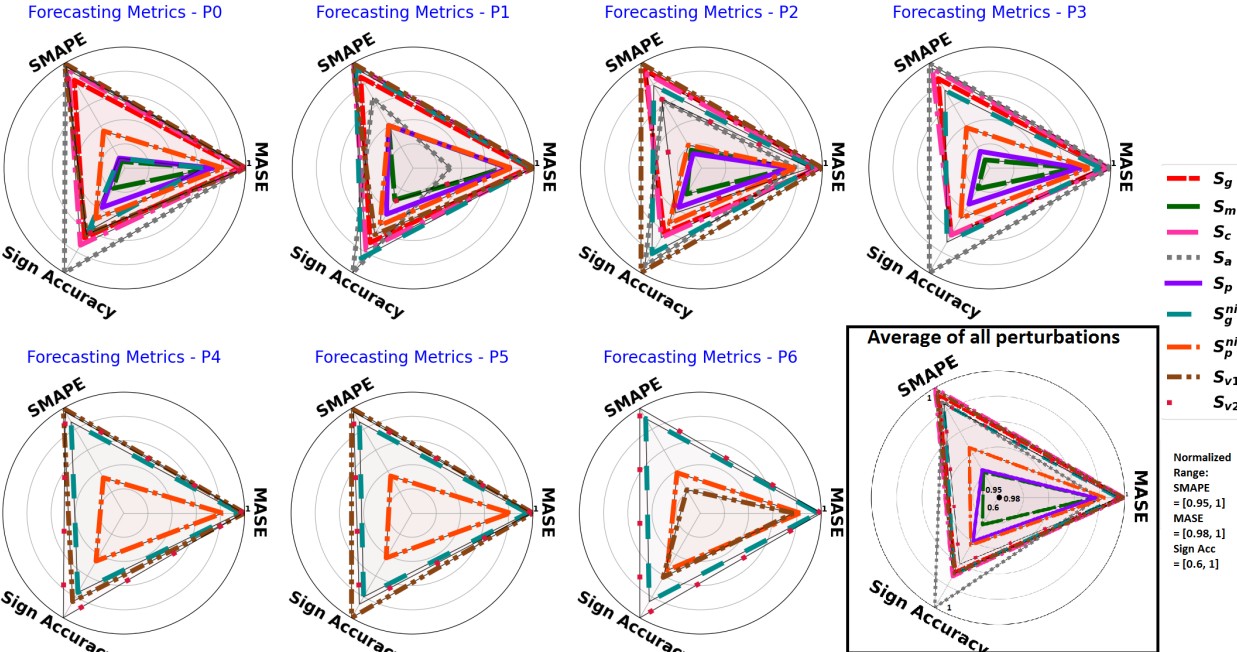

Figure 7: Radar plots showing accuracy metrics for all models under different perturbations (P1 through P6) and their average (bottom right). **Lower values (i.e., lines closer to the center) indicate lower accuracy, while points farther from the center represent better accuracy across metrics.**

| Forecasting Evaluation Dimensions | P | Partial Order | Complete Order |
|---|---|---|---|
| Inter-industry statistical bias ($WRS_I\downarrow$) | P0 | {$S_g$: 4.6, $S_m$: 4.6, $S_{v2}$: 4.6, $S_r$: 4.6, $S_c$: 5.9, $S_a$: 5.9, $S_p^{ni}$: 5.9, $S_{v1}$: 5.9, $S_g^{ni}$: 6.9, $S_p$: 6.9, $S_b$: 6.9 } | {$S_g$: 1, $S_m$: 1, $S_{v2}$: 1, $S_r$: 1, $S_c$: 2, $S_a$: 2, $S_p^{ni}$: 2, $S_{v1}$: 2, $S_g^{ni}$: 3, $S_p$: 3, $S_b$: 3 } |
| | P1 | {$S_a$: 2.6, $S_m$: 4.6, $S_g$: 4.6, $S_g^{ni}$: 4.6, $S_r$: 4.6, $S_p^{ni}$: 5.9, $S_{v1}$: 5.9, $S_p$: 6.9, $S_c$: 6.9, $S_{v2}$: 6.9, $S_b$: 6.9 } | {$S_a$: 1, $S_m$: 1, $S_g$: 1, $S_g^{ni}$: 1, $S_r$: 1, $S_p^{ni}$: 2, $S_{v1}$: 2, $S_p$: 3, $S_c$: 3, $\tilde{S}_{v2}$: 3, $S_{v2}$: 3, $S_b$: 3} |
| | P2 | {$S_a$: 4.6, $S_g$: 4.6, $S_g^{ni}$: 4.6, $S_p^{ni}$: 4.6, $S_m$: 4.6, $S_r$: 4.6, $S_{v1}$: 5.9, $S_c$: 6.9, $S_p$: 6.9, $S_{v2}$: 6.9, $S_b$: 6.9} | {$S_a$: 1, $S_g$: 1, $S_g^{ni}$: 1, $S_p^{ni}$: 1, $S_m$: 1, $S_r$: 1, $S_{v1}$: 2, $S_c$: 3, $S_p$: 3, $S_{v2}$: 3, $S_b$: 3} |
| | P3 | {$S_g$: 4.6, $S_g^{ni}$: 4.6, $S_m$: 4.6, $S_r$: 4.6, $S_c$: 4.6, $S_a$: 5.9, $S_p^{ni}$: 6.9, $S_p$: 6.9, $S_b$: 6.9} | {$S_g$: 1, $S_g^{ni}$: 1, $S_m$: 1, $S_r$: 1, $S_c$: 1, $S_a$: 2, $S_p$: 3, $S_p^{ni}$: 3, $S_b$: 3} |
| | P4 | {$S_{v2}$: 4.6, $S_g^{ni}$: 4.6, $S_r$: 4.6, $S_{v1}$: 5.9, $S_p^{ni}$: 5.9, $S_b$: 6.9 } | {$S_{v2}$: 1, $S_g^{ni}$: 1, $S_r$: 1, $S_{v1}$: 2, $S_p^{ni}$: 2, $S_b$: 3 } |
| | P5 | {$S_{v2}$: 4.6, $S_g^{ni}$: 4.6, $S_r$: 5.2, $S_{v1}$: 5.9, $S_p^{ni}$: 5.9, $S_b$: 6.9} | {$S_{v2}$: 1, $S_g^{ni}$: 1, $S_r$: 1, $S_{v1}$: 2, $S_p^{ni}$: 2, $S_b$: 3} |
| | P6 | {$S_r$: 4.6, $S_g^{ni}$: 4.6, $S_{v2}$: 5.2, $S_{v1}$: 5.9, $S_p^{ni}$: 5.9, $S_b$: 6.9} | {$S_r$: 1, $S_g^{ni}$: 1, $S_{v2}$: 1, $S_{v1}$: 2, $S_p^{ni}$: 2, $S_b$: 2} |
| Intra-industry statistical bias ($WRS_C\downarrow$) | P0 | {$S_a$: 2.6, $S_{v2}$: 4.6, $S_g$: 4.6, $S_g^{ni}$: 4.6, $S_p^{ni}$: 4.6, $S_r$: 4.6, $S_c$: 5.6, $S_{v1}$: 5.9, $S_p$: 6.9, $S_m$: 6.9, $S_r$: 6.9, $S_b$: 6.9 } | {$S_a$: 1, $S_g$: 1, $S_g^{ni}$: 1, $S_p^{ni}$: 1, $S_r$: 1, $S_c$: 2, $S_{v1}$: 3, $S_p$: 3, $S_m$: 3, $S_r$: 3, $S_b$: 3} |
| | P1 | {$S_a$: 0.6, $S_c$: 4.6, $S_{v1}$: 4.6, $S_{v2}$: 4.6, $S_p$: 5.9, $S_p^{ni}$: 5.9, $S_g^{ni}$: 5.9, $S_r$: 5.9, $S_g$: 6.9, $S_m$: 6.9, $S_b$: 6.9 } | {$S_a$: 1, $S_c$: 1, $S_{v1}$: 1, $S_{v2}$: 1, $S_p$: 2, $S_p^{ni}$: 2, $S_g^{ni}$: 2, $S_r$: 2, $S_g$: 3, $S_m$: 3, $S_b$: 3 } |
| | P2 | {$S_a$: 2.6, $S_c$: 4.6, $S_p^{ni}$: 4.6, $S_{v1}$: 4.6, $S_r$: 4.6, $S_{v2}$: 5.2, $S_p$: 5.2, $S_g$: 5.9, $S_g^{ni}$: 5.9, $S_m$: 6.9, $S_b$: 6.9 } | {$S_a$: 1, $S_c$: 1, $S_p^{ni}$: 1, $S_{v1}$: 1, $S_r$: 1, $S_{v2}$: 2, $S_p$: 2, $S_g$: 2, $S_g^{ni}$: 2, $S_m$: 3, $S_b$: 3 } |
| | P3 | {$S_g$: 4.6, $S_g^{ni}$: 4.6, $S_p^{ni}$: 4.6, $S_p$: 4.6, $S_c$: 4.6, $S_m$: 6.9, $S_a$: 6.9, $S_r$: 6.9, $S_b$: 6.9} | {$S_g$: 1, $S_g^{ni}$: 1, $S_p^{ni}$: 1, $S_p$: 1, $S_c$: 1, $S_m$: 2, $S_a$: 2, $S_r$: 2, $S_b$: 2} |
| | P4 | {$S_{v2}$: 4.6, $S_{v1}$: 5.9, $S_p^{ni}$: 6.9, $S_g^{ni}$: 6.9, $S_r$: 6.9, $S_b$: 6.9 } | {$S_{v2}$: 1, $S_{v1}$: 2, $S_p^{ni}$: 3, $S_g^{ni}$: 3, $S_r$: 3, $S_b$: 3 } |
| | P5 | {$S_{v2}$: 4.6, $S_p^{ni}$: 4.6, $S_{v1}$: 5.2, $S_r$: 5.9, $S_g^{ni}$: 6.9, $S_b$: 6.9 } | {$S_{v2}$: 1, $S_p^{ni}$: 1, $S_{v1}$: 1, $S_r$: 2, $S_g^{ni}$: 3, $S_b$: 3 } |
| | P6 | {$S_r$: 4.6, $S_p^{ni}$: 4.6, $S_{v2}$: 5.9, $S_{v1}$: 6.9, $S_g^{ni}$: 6.9, $S_b$: 6.9 } | {$S_r$: 1, $S_p^{ni}$: 1, $S_{v2}$: 2, $S_{v1}$: 3, $S_g^{ni}$: 3, $S_b$: 3 } |
| Confounding Bias with *Industry* as confounder ($PIE_I\%\downarrow$) | P1 | {$S_g^{ni}$: 437.28, $S_{v1}$: 539.98, $S_{v2}$: 609.44, $S_g$: 1012.79, $S_a$: 1223.59, $S_p$: 1586.51, $S_p^{ni}$: 2003.93, $S_r$: 4109.55, $S_c$: 4111.33, $S_m$: 4176.43, $S_b$: 5237.52 } | {$S_g^{ni}$: 1, $S_{v1}$: 1, $S_{v2}$: 1, $S_g$: 1, $S_a$: 2, $S_p$: 2, $S_p^{ni}$: 2, $S_r$: 2, $S_c$: 3, $S_m$: 3, $S_b$: 3 } |
| | P2 | {$S_{v2}$: 539.12, $S_a$: 1531.37, $S_g^{ni}$: 2086.5, $S_p^{ni}$: 2211.66, $S_m$: 2536.07, $S_p$: 3071.64, $S_g$: 3140.33, $S_c$: 3909.98, $S_{v1}$: 4405.36, $S_r$: 4681.26, $S_b$: 9593.46} | {$S_{v2}$: 1, $S_a$: 1, $S_g^{ni}$: 1, $S_p^{ni}$: 1, $S_m$: 2, $S_p$: 2, $S_g$: 2, $S_c$: 2, $S_{v1}$: 3, $S_r$: 3, $S_b$: 3} |
| | P3 | {$S_g$: 349.16, $S_g^{ni}$: 493.42, $S_c$: 1036.66, $S_p^{ni}$: 1247.39, $S_a$: 2251.9, $S_p$: 3043.22, $S_r$: 4509.75, $S_b$: 5225.69, $S_m$: 5715.21} | {$S_g$: 1, $S_g^{ni}$: 1, $S_c$: 1, $S_p^{ni}$: 2, $S_a$: 2, $S_p$: 2, $S_r$: 3, $S_b$: 3, $S_m$: 3} |
| | P4 | {$S_{v1}$: 450.09, $S_p^{ni}$: 1372.59, $S_g^{ni}$: 1498.45, $S_{v2}$: 1941.72, $S_r$: 3916.46, $S_b$: 6746.66} | {$S_{v1}$: 1, $S_p^{ni}$: 1, $S_g^{ni}$: 2, $S_{v2}$: 2, $S_r$: 3, $S_b$: 3} |
| | P5 | {$S_g^{ni}$: 2391.58, $S_{v1}$: 2571.56, $S_{v2}$: 4128.95, $S_r$: 5004.2, $S_b$: 7883.56, $S_p^{ni}$: 13472.24} | {$S_g^{ni}$: 1, $S_{v1}$: 1, $S_{v2}$: 2, $S_r$: 2, $S_b$: 3, $S_p^{ni}$: 3} |
| | P6 | {$S_{v2}$: 205.36, $S_g^{ni}$: 789.78, $S_p^{ni}$: 1687.56, $S_r$: 2586.99, $S_{v1}$: 4215.45, $S_b$: 9606.06} | {$S_{v2}$: 1, $S_g^{ni}$: 1, $S_p^{ni}$: 2, $S_r$: 2, $S_{v1}$: 3, $S_b$: 3} |
| Confounding Bias with *Company* as confounder ($PIE_C\%\downarrow$) | P1 | {$S_g^{ni}$: 632.31, $S_a$: 647.15, $S_{v2}$: 761.01, $S_g$: 882.66, $S_m$: 910.47, $S_p$: 914.62, $S_p^{ni}$: 993.88, $S_r$: 1460.55, $S_c$: 1630.63, $S_b$: 1855.99, $S_{v1}$: 1923.8} | {$S_g^{ni}$: 1, $S_a$: 1, $S_{v2}$: 1, $S_g$: 1 $S_m$: 2, $S_p$: 2, $S_p^{ni}$: 2, $S_r$: 2, $S_c$: 3, $S_b$: 3, $S_{v1}$: 3 } |
| | P2 | {$S_r$: 295.4, $S_{v1}$: 516.73, $S_g^{ni}$: 709.9, $S_g$: 829.27, $S_{v2}$: 984.88, $S_m$: 1119.67, $S_a$: 1234.68, $S_b$: 2132.45, $S_p$: 2566.84, $S_p^{ni}$: 4767.08, $S_c$: 4963.77} | {$S_r$: 1, $S_{v1}$: 1, $S_g^{ni}$: 1, $S_g$: 1, $S_{v2}$: 2, $S_m$: 2, $S_a$: 2, $S_b$: 2, $S_p$: 3, $S_p^{ni}$: 3, $S_c$: 3} |
| | P3 | {$S_p^{ni}$: 429.43, $S_g$: 476.25, $S_m$: 807.34, $S_c$: 880.64, $S_g^{ni}$: 938.41, $S_a$: 1801.04, $S_r$: 2051.79, $S_b$: 2201.23, $S_p$: 3572.98} | {$S_p^{ni}$: 1, $S_g$: 1, $S_m$: 1, $S_c$: 2, $S_g^{ni}$: 2, $S_a$: 2, $S_r$: 3, $S_b$: 3, $S_p$: 3} |
| | P4 | {$S_g^{ni}$: 363.8, $S_p^{ni}$: 760.76, $S_r$: 1038.87, $S_{v2}$: 1584.25, $S_{v1}$: 2122.6, $S_b$: 2635.14} | {$S_g^{ni}$: 1, $S_p^{ni}$: 1, $S_r$: 2, $S_{v2}$: 2, $S_{v1}$: 3, $S_b$: 3} |
| | P5 | {$S_g^{ni}$: 960.09, $S_{v1}$: 1058.15, $S_r$: 1211.5, $S_p^{ni}$: 1522.82, $S_{v2}$: 1596.87, $S_b$: 1952.83} | {$S_g^{ni}$: 1, $S_{v1}$: 1, $S_r$: 2, $S_p^{ni}$: 2, $S_{v2}$: 3, $S_b$: 3} |
| | P6 | {$S_{v1}$: 678.24, $S_{v2}$: 1112.99, $S_r$: 1155.87, $S_g^{ni}$: 1990.4, $S_b$: 2027.3, $S_p^{ni}$: 2867.36} | {$S_{v1}$: 1, $S_{v2}$: 1, $S_r$: 2, $S_g^{ni}$: 2, $S_b$: 3, $S_p^{ni}$: 3} |
| Perturbation Impact with *Industry* as the confounder ($APE_I\downarrow$) | P1 | {$S_g^{ni}$: 5.02, $S_{v1}$: 5.42, $S_c$: 8.84, $S_m$: 10.86, $S_g$: 14.52, $S_{v2}$: 19.25, $S_p$: 24.44, $S_p^{ni}$: 25.11, $S_r$: 42.84, $S_b$: 51.0, $S_a$: 87.51} | {$S_g^{ni}$: 1, $S_{v1}$: 1, $S_c$: 1, $S_m$: 1, $S_g$: 2, $S_{v2}$: 2, $S_p$: 2, $S_p^{ni}$: 2, $S_r$: 3, $S_b$: 3, $S_a$: 3} |
| | P2 | {$S_g$: 2.69, $S_{v1}$: 3.77, $S_p^{ni}$: 4.95, $S_{v2}$: 6.04, $S_g^{ni}$: 9.12, $S_c$: 9.54, $S_p$: 13.09, $S_a$: 16.38, $S_m$: 16.58, $S_r$: 25.52, $S_b$: 101.14 } | {$S_g$: 1, $S_{v1}$: 1, $S_p^{ni}$: 1, $S_{v2}$: 1, $S_g^{ni}$: 2, $S_c$: 2, $S_p$: 2, $S_a$: 2, $S_m$: 3, $S_r$: 3, $S_b$: 3 } |
| | P3 | {$S_m$: 3.6, $S_g$: 4.88, $S_g^{ni}$: 5.31, $S_a$: 7.5, $S_c$: 11.51, $S_p$: 14.8, $S_r$: 15.0, $S_p^{ni}$: 16.25, $S_b$: 51.42} | {$S_m$: 1, $S_g$: 1, $S_g^{ni}$: 1, $S_a$: 2, $S_c$: 2, $S_p$: 2, $S_r$: 3, $S_p^{ni}$: 3, $S_b$: 3} |
| | P4 | {$S_{v2}$: 4.67, $S_g^{ni}$: 5.21, $S_{v1}$: 6.38, $S_p^{ni}$: 16.06, $S_r$: 41.37, $S_b$: 53.09} | {$S_{v2}$: 1, $S_g^{ni}$: 1, $S_{v1}$: 2, $S_p^{ni}$: 2, $S_r$: 3, $S_b$: 3} |
| | P5 | {$S_g^{ni}$: 4.6, $S_{v2}$: 5.24, $S_{v1}$: 7.9, $S_p^{ni}$: 10.74, $S_r$: 55.88, $S_b$: 98.69} | {$S_g^{ni}$: 1, $S_{v2}$: 1, $S_{v1}$: 2, $S_p^{ni}$: 2, $S_r$: 3, $S_b$: 3} |
| | P6 | {$S_{v2}$: 1.06, $S_g^{ni}$: 1.15, $S_{v1}$: 9.15, $S_p^{ni}$: 18.32, $S_r$: 27.08, $S_b$: 52.01} | {$S_{v2}$: 1, $S_g^{ni}$: 1, $S_{v1}$: 2, $S_p^{ni}$: 2, $S_r$: 3, $S_b$: 3} |
| Perturbation Impact with *Company* as the confounder ($APE_C\downarrow$) | P1 | {$S_b$: 0.0, $S_{v1}$: 5.47, $S_g^{ni}$: 6.46, $S_g$: 6.86, $S_c$: 8.2, $S_r$: 11.38, $S_m$: 11.78, $S_{v2}$: 17.26, $S_p$: 17.99, $S_p^{ni}$: 25.97, $S_a$: 39.23} | {$S_b$: 1, $S_{v1}$: 1, $S_g^{ni}$: 1, $S_g$: 1, $S_c$: 2, $S_r$: 2, $S_m$: 2, $S_{v2}$: 2, $S_p$: 3, $S_p^{ni}$: 3, $S_a$: 3} |
| | P2 | {$S_b$: 0.0, $S_p^{ni}$: 1.94, $S_c$: 3.94, $S_{v2}$: 3.94, $S_g$: 4.33, $S_{v1}$: 6.14, $S_r$: 6.17, $S_m$: 7.78, $S_g^{ni}$: 8.14, $S_a$: 13.25, $S_p$: 19.38 } | {$S_b$: 1, $S_p^{ni}$: 1, $S_c$: 1, $S_{v2}$: 1, $S_g$: 2, $S_{v1}$: 2, $S_r$: 2, $S_m$: 2, $S_g^{ni}$: 3, $S_a$: 3, $S_p$: 3 } |
| | P3 | {$S_b$: 0.0, $S_a$: 2.5, $S_c$: 4.78, $S_g$: 5.48, $S_r$: 6.13, $S_p$: 6.17, $S_p^{ni}$: 6.25, $S_m$: 8.36, $S_g^{ni}$: 9.3} | {$S_b$: 1, $S_a$: 1, $S_c$: 1, $S_g$: 2, $S_r$: 2, $S_p$: 2, $S_p^{ni}$: 3, $S_m$: 3, $S_g^{ni}$: 3} |
| | P4 | {$S_b$: 0.0, $S_{v2}$: 2.11, $S_{v1}$: 3.36, $S_g^{ni}$: 5.5, $S_r$: 9.49, $S_p^{ni}$: 11.09} | {$S_b$: 1, $S_{v2}$: 1, $S_{v1}$: 2, $S_g^{ni}$: 2, $S_r$: 3 $S_p^{ni}$: 3} |
| | P5 | {$S_b$: 0.0, $S_{v2}$: 3.96, $S_{v1}$: 5.52, $S_g^{ni}$: 6.67, $S_p^{ni}$: 7.65, $S_r$: 13.53} | {$S_b$: 1, $S_{v2}$: 1, $S_{v1}$: 2, $S_g^{ni}$: 2, $S_p^{ni}$: 3, $S_r$: 3} |
| | P6 | {$S_b$: 0.0, $S_{v2}$: 1.03, $S_g^{ni}$: 4.18, $S_r$: 4.49, $S_p^{ni}$: 7.59, $S_{v1}$: 20.31} | {$S_b$: 1, $S_{v2}$: 1, $S_g^{ni}$: 2, $S_r$: 2, $S_p^{ni}$: 3, $S_{v1}$: 3} |

Table 3: Final raw scores and ratings based on different metrics computed. Higher rating indicate higher bias.

| Forecasting Evaluation Dimensions | P | Partial Order | Complete Order |
|---|---|---|---|
| Accuracy (SMAPE↓) | P0 | $\{S_{v1}: 0.039, S_a: 0.040, S_{v2}: 0.041, S_c: 0.043, S_g: 0.049, S_p^{ni}: 0.079, S_p: 0.095, S_g^{ni}: 0.095, S_m: 0.097, S_r: 0.829, S_b: 1.276 \}$ | $\{S_{v1}: 1, S_a: 1, S_{v2}: 1, S_c: 1, S_g: 2, S_p^{ni}: 2, S_p: 2, S_g^{ni}: 2, S_m: 3, S_r: 3, S_b: 3 \}$ |
| | P1 | $\{S_{v1}: 0.064, S_c: 0.065, S_g^{ni}: 0.067, S_g: 0.072, S_a: 0.084, S_m: 0.100, S_p: 0.100, S_p^{ni}: 0.100, S_{v2}: 0.127, S_r: 0.830, S_b: 1.276 \}$ | $\{S_{v1}: 1, S_c: 1, S_g^{ni}: 1, S_g: 2, S_a: 2, S_m: 2, S_p: 2, S_p^{ni}: 2, S_{v2}: 3, S_r: 3, S_b: 3 \}$ |
| | P2 | $\{S_{v1}: 0.047, S_g: 0.051, S_c: 0.053, S_g^{ni}: 0.060, S_{v2}: 0.068, S_a: 0.069, S_p^{ni}: 0.095, S_m: 0.098, S_p: 0.100, S_r: 0.830, S_b: 1.276 \}$ | $\{S_{v1}: 1, S_g: 1, S_c: 1, S_g^{ni}: 1, S_{v2}: 2, S_a: 2, S_p^{ni}: 2, S_m: 2, S_p: 3, S_r: 3, S_b: 3 \}$ |
| | P3 | $\{S_a: 0.040, S_c: 0.043, S_g: 0.049, S_g^{ni}: 0.056, S_p^{ni}: 0.078, S_p: 0.092, S_m: 0.097, S_r: 0.830, S_b: 1.276 \}$ | $\{S_a: 1, S_c: 1, S_g: 1, S_g^{ni}: 2, S_p^{ni}: 2, S_p: 2, S_m: 3, S_r: 3, S_b: 3 \}$ |
| | P4 | $\{S_{v1}: 0.039, S_{v2}: 0.041, S_g^{ni}: 0.047, S_p^{ni}: 0.080, S_r: 0.830, S_b: 1.276 \}$ | $\{S_{v1}: 1, S_{v2}: 1, S_g^{ni}: 2, S_p^{ni}: 2, S_r: 3, S_b: 3 \}$ |
| | P5 | $\{S_{v1}: 0.039, S_{v2}: 0.041, S_g^{ni}: 0.047, S_p^{ni}: 0.079, S_r: 0.829, S_b: 1.276 \}$ | $\{S_{v1}: 1, S_{v2}: 1, S_g^{ni}: 2, S_p^{ni}: 2, S_r: 3, S_b: 3 \}$ |
| | P6 | $\{S_{v2}: 0.041, S_g^{ni}: 0.047, S_p^{ni}: 0.079, S_{v1}: 0.089, S_r: 0.832, S_b: 1.276 \}$ | $\{S_{v2}: 1, S_g^{ni}: 1, S_p^{ni}: 2, S_{v1}: 2, S_r: 3, S_b: 3 \}$ |
| Accuracy (MASE↓) | P0 | $\{S_{v1}: 3.68, S_a: 3.79, S_{v2}: 3.89, S_c: 4.18, S_g: 4.64, S_p^{ni}: 7.19, S_p: 8.91, S_m: 9.03, S_g^{ni}: 10.37, S_r: 86.45, S_b: 947.56\}$ | $\{S_{v1}: 1, S_a: 1, S_{v2}: 1, S_c: 1, S_g: 2, S_p^{ni}: 2, S_p: 2, S_m: 2, S_g^{ni}: 3, S_r: 3, S_b: 3\}$ |
| | P1 | $\{S_{v1}: 5.30, S_c: 5.40, S_g^{ni}: 5.65, S_g: 6.13, S_p^{ni}: 8.87, S_p: 9.19, S_m: 9.32, S_{v2}: 11.18, S_a: 18.36, S_r: 86.99, S_b: 947.56\}$ | $\{S_{v1}: 1, S_c: 1, S_g^{ni}: 1, S_g: 1, S_p^{ni}: 2, S_p: 2, S_m: 2, S_{v2}: 2, S_a: 3, S_r: 3, S_b: 3\}$ |
| | P2 | $\{S_{v1}: 4.24, S_g: 4.74, S_c: 4.99, S_g^{ni}: 5.59, S_{v2}: 6.16, S_a: 8.24, S_p^{ni}: 8.49, S_m: 9.15, S_p: 9.32, S_r: 86.87, S_b: 947.56\}$ | $\{S_{v1}: 1, S_g: 1, S_c: 1, S_g^{ni}: 1, S_{v2}: 2, S_a: 2, S_p^{ni}: 2, S_m: 2, S_p: 3, S_r: 3, S_b: 3\}$ |
| | P3 | $\{S_a: 3.79, S_c: 4.10, S_g: 4.64, S_g^{ni}: 5.39, S_p^{ni}: 7.11, S_p: 8.68, S_m: 9.03, S_r: 86.65, S_b: 947.56 \}$ | $\{S_a: 1, S_c: 1, S_g: 1, S_g^{ni}: 2, S_p^{ni}: 2, S_p: 2, S_m: 3, S_r: 3, S_b: 3 \}$ |
| | P4 | $\{S_{v1}: 3.68, S_{v2}: 3.89, S_g^{ni}: 4.46, S_p^{ni}: 7.10, S_r: 86.65, S_b: 947.56\}$ | $\{S_{v1}: 1, S_{v2}: 1, S_g^{ni}: 2, S_p^{ni}: 2, S_r: 3, S_b: 3\}$ |
| | P5 | $\{S_{v1}: 3.67, S_{v2}: 3.90, S_g^{ni}: 4.47, S_p^{ni}: 7.23, S_r: 86.53, S_b: 947.56\}$ | $\{S_{v1}: 1, S_{v2}: 1, S_g^{ni}: 2, S_p^{ni}: 2, S_r: 3, S_b: 3\}$ |
| | P6 | $\{S_{v2}: 3.93, S_g^{ni}: 4.42, S_p^{ni}: 7.24, S_{v1}: 8.26, S_r: 87.20, S_b: 947.56\}$ | $\{S_{v2}: 1, S_g^{ni}: 1, S_p^{ni}: 2, S_{v1}: 2, S_r: 3, S_b: 3\}$ |
| Accuracy (Sign Accuracy %↑) | P0 | $\{S_m: 40.70, S_p: 45.09, S_p^{ni}: 47.67, S_r: 49.88, S_g^{ni}: 50.41, S_{v2}: 51.28, S_{v1}: 51.32, S_g: 52.08, S_c: 53.75, S_a: 60.08, S_b: 62.60 \}$ | $\{S_m: 1, S_p: 1, S_p^{ni}: 1, S_r: 1, S_g^{ni}: 2, S_{v2}: 2, S_{v1}: 2, S_g: 2, S_c: 3, S_a: 3, S_b: 3 \}$ |
| | P1 | $\{S_m: 41.19, S_{v2}: 41.54, S_p: 44.33, S_p^{ni}: 46.77, S_{v1}: 48.77, S_r: 49.62, S_g: 50.53, S_c: 52.09, S_g^{ni}: 53.93, S_a: 57.08, S_b: 62.60 \}$ | $\{S_m: 1, S_{v2}: 1, S_p: 1, S_p^{ni}: 1, S_{v1}: 2, S_r: 2, S_g: 2, S_c: 2, S_g^{ni}: 3, S_a: 3, S_b: 3 \}$ |
| | P2 | $\{S_m: 41.05, S_p: 44.02, S_{v2}: 45.28, S_p^{ni}: 47.67, S_r: 49.64, S_g: 49.75, S_c: 50.79, S_g^{ni}: 54.43, S_a: 57.13, S_{v1}: 58.69, S_b: 62.60 \}$ | $\{S_m: 1, S_p: 1, S_{v2}: 1, S_p^{ni}: 1, S_r: 2, S_g: 2, S_c: 2, S_g^{ni}: 2, S_a: 3, S_{v1}: 3, S_b: 3 \}$ |
| | P3 | $\{S_m: 40.72, S_p: 44.26, S_p^{ni}: 47.50, S_r: 49.71, S_g: 51.34, S_c: 51.35, S_g^{ni}: 52.97, S_a: 59.98, S_b: 62.60 \}$ | $\{S_m: 1, S_p: 1, S_p^{ni}: 1, S_r: 2, S_g: 2, S_c: 2, S_g^{ni}: 3, S_a: 3, S_b: 3 \}$ |
| | P4 | $\{S_p^{ni}: 42.87, S_r: 49.71, S_g^{ni}: 49.46, S_{v1}: 51.35, S_{v2}: 54.74, S_b: 62.60 \}$ | $\{S_p^{ni}: 1, S_r: 1, S_g^{ni}: 2, S_{v1}: 2, S_{v2}: 3, S_b: 3 \}$ |
| | P5 | $\{S_p^{ni}: 41.60, S_g^{ni}: 49.63, S_r: 49.67, S_{v2}: 51.14, S_{v1}: 53.95, S_b: 62.60 \}$ | $\{S_p^{ni}: 1, S_g^{ni}: 1, S_r: 2, S_{v2}: 2, S_{v1}: 3, S_b: 3 \}$ |
| | P6 | $\{S_p^{ni}: 42.60, S_g^{ni}: 48.78, S_{v1}: 43.97, S_r: 50.05, S_{v2}: 52, S_b: 62.60 \}$ | $\{S_p^{ni}: 1, S_g^{ni}: 1, S_{v1}: 2, S_r: 2, S_{v2}: 3, S_b: 3\}$ |

Table 4: Final raw scores and ratings based on different metrics computed. Higher rating indicate higher inaccuracy. For simplicity, we denoted the raw scores for accuracy metrics using just the mean value, but standard deviation was also considered for rating. The chosen rating level, L = 3.

| Metric | Q1 | Q2 | Q4 | Q5 | Q6 | Q8 | Q9 | Q10 | Q12 | Q13 | Q14 |
|---|---|---|---|---|---|---|---|---|---|---|---|
| $\mu$ | 3.1923 | 2.8077 | 2.5385 | 2.7692 | 2.9231 | 2.6923 | 2.9231 | 3.2308 | 2.6538 | 2.8077 | 3.0769 |
| $\sigma$ | 1.2335 | 1.3570 | 1.3336 | 1.1767 | 1.3834 | 1.0870 | 1.2625 | 1.4507 | 1.1981 | 1.3570 | 1.4676 |
| t-statistic | 4.9287 | 3.0349 | 2.0588 | 3.3333 | 3.4023 | 3.2476 | 3.7282 | 4.3259 | 2.7828 | 3.0349 | 3.7417 |
| p-value | 0.0000* | 0.0028* | 0.0250* | 0.0013* | 0.0011* | 0.0017* | 0.0005* | 0.0001* | 0.0051* | 0.0028* | 0.0005* |

Table 5: Summary of one sample right-tailed t-test results: Comparison of sample means to the hypothesized mean of 2 with a sample size of 26. The right-tailed p-values indicate whether the sample means are significantly greater the hypothesized mean. * denotes that mean of responses for all the questions is greater than 2.

| Hypothesis | Test Performed | Statistics | Conclusion |
|---|---|---|---|
| There is a high positive correlation between users' fairness rankings and rankings generated by our rating method. | Spearman Rank Correlation | $\rho = 0.73$ | The fairness rankings generated by our rating method aligns well with users' rankings. |
| The mean of the responses for Q4 is less than or equal to the mean of the responses for Q6. | Paired t-test | t-statistic: -1.18, p-val: 0.12 | Users found it easy to interpret the behavior of the systems from rankings compared to graphs and statistics with a confidence interval of 85 %. |
| There is a very high positive correlation between users' rankings and rankings generated by our rating method. | Spearman Rank Correlation | $\rho$: 0.91 | The robustness rankings generated by our rating method aligns very well with users' rankings. |
| The mean of the responses for Q8 is less than or equal to the mean of the responses for Q10. | Paired t-test | t-statistic: -1.89, p-val: 0.03 | Users found it easy to interpret the behavior of the systems from rankings compared to graphs and statistics with a confidence interval of 95 %. |
| There is a weak positive correlation between users' rankings and rankings generated by our rating method. | Spearman Rank Correlation | $\rho$: 0.14 | The robustness rankings generated by our rating method weakly aligns with users' rankings. |
| The mean of the responses for Q12 is less than or equal to the mean of the responses for Q14. | Paired t-test | t-statistic: -1.62, p-val: 0.06 | Users found it easy to interpret the behavior of the systems from rankings compared to graphs and statistics with a confidence interval of 90 %. |

Table 6: Table with the hypotheses evaluated in the user study, statistical tests used to validate the hypotheses, results obtained, and conclusions drawn.

