# OpenReview forum: "Creating a Causally Grounded Rating Method for Assessing the Robustness of AI Models for Time-Series Forecasting"
_TMLR — Under review for TMLR_

### Review · Reviewer_K55g · 2026-07-01

**Summary Of Contributions:**

This paper presents a causally grounded rating framework for evaluating the robustness of time-series forecasting models (TSFM). It extends multiple prior works, including evaluating two ViT-num-spec models proposed in Zeng et al. (2023), evaluating multimodal models and multimodal attacks beyond prior work in finance-domain perturbation, and extending Lakkaraju et al. for rating TSFM for robustness against perturbations. The research questions are clear; they focus on comparing foundation and non-foundation models, unimodal and multimodal models, and different model architectures. This paper defines 6 perturbations: 2 semantic (drop-to-zero and value-halved), 1 syntactic (missing values), 2 input-specific (single pixel change and saturation change), and 1 composite (TSFM combination with another AI system). There is also a comprehensive suite of evaluation metrics, including robustness and forecasting accuracy. The pipeline has two parts: one from data to predictions, and the other from predictions to ratings. It evaluates 5 types of foundation models and 3 additional baselines using one year of stock price data from six leading companies across three industries. In the experimental results, the paper shows that 1) sensitive attribute affects the residual, even though sensitive attribute has no effect on perturbation; 2) confounder affects the relationship between perturbation and residual, particularly when the confounder influences the perturbation; 3) perturbation affects the residual when sensitive attribute may have an effect on residual, and 4) perturbation degrades the accuracy of rating set. Results also show that multimodal TSFM is more robust, whereas unimodal TSFM achieves higher forecasting accuracy. TSFM, trained on time-series tasks, achieves greater robustness and accuracy than general-purpose models. Additionally, a user study (n=26) validates the effectiveness of the ratings.

Key strengths:
1. The overall pipeline and implementation are clear, thus increasing reproducibility.
2. Evaluating TSFMs for their robustness is a very important research problem in ML for finance.
3. Very comprehensive evaluation metrics and evaluated models.

Key weaknesses:
1. Under Table 2: "Snig : 5.89, Sv1: 7.34, Sc :7.80, Sm : 9.83, Sg : 6.46,Sv1 : 6.46, Snip : 12.66, Sp :15.98, Sr : 21.57, Sa : 27.73,Sb : 33.9", there are two Sv1 with different values.
2. The Propensity Score Matching PIE score is not correctly ranged within 0-1, and scaling it by 100 does not make sense to use it as a percentage score. This definition seems problematic.
3. The ViT-num-spec model was trained using two real-world datasets; Sv1 was trained on S&P 500 stock data from 2000-2014, and Sv2 was trained on data from March 2020–November 2022. However, the MOMENT model is used in zero-shot and few-shot settings. The comparison between the two is unfair.

**Audience:**

Yes

**Audience Explanation:**

ML researchers in finance will be interested in the paper's findings, especially those working on time-series forecasting and foundation models for finance.

**Broader Impact Concerns:**

No concern about the ethical implications of this work.

**Claims And Evidence:**

No

**Claims Explanation:**

There are some key weaknesses that make the claims poorly supported. Specifically, some results have typos or errors, making them less trustworthy. The metric definition is also problematic, and comparisons between models are not fair.

**Requested Changes:**

1. Check the results again to see if there are typos and errors in them. If it's not an error, please explain why there are two different values for the same model.
2. Can you explain why $||APE_o| - |APE_m||$ is $[0, 1]$. If not, scaling it with 100 doesn't make sense. The authors should revise the propensity score matching.
3. Add an explanation for why directly comparing fine-tuned models with zero-shot models is a fair comparison. This does not lead to the conclusion that A is better than B. If it's not fair, the experiments need to be redesigned and the claims in the paper updated.
4. The "large-scale experimental setup" is a bit of an overclaim because the evaluation datasets only have one year of stock price and a limited number of companies. Consider changing the wording.
5. (minor) The whole paper's experimental results are not surprising, which is fine, but it is better to revise the paper to emphasize why these research questions are important in this field.

---

### Review · Reviewer_7ofw · 2026-07-21

**Summary Of Contributions:**

The paper proposes a framework that is causally grounded for time-series forecasting robustness rating. The robustness described in the paper primarily emphasizes the perturbation of the time-series inputs. The framework integrates statistics and causality to evaluate the effects of perturbations, sensitive attributes, and confounding variables on residuals, with additional metrics for robustness and accuracy. The paper evaluates the framework on financial time-series forecasting and compares various baselines, including time-series foundation models, general-purpose foundation models, and multimodal models. An user study is also conducted to evaluate user experience on how well the framework identifies robustness and fairness.

Strengths:
- The defined metrics are non-trivial in identifying the weaknesses of existing time-series foundation models.
- The paper is structurally organized and the experimental workflow is relatively easy to follow.
- The user study is helpful in illustrating the impact of the work.

Several major concerns I have limit the impact of the paper:
- My biggest concern is that the robustness considered primarily relies on a small collection of manually designed input perturbations. However, in financial fields, a major source of failure is domain and distribution shift rather than signal level corruption. Some real-world challenges include changes in market regimes across various time-scales, volatility, and many other correlations. This limits the practicality of the work without more realistic forms of perturbation for robustness evaluation.
- I am concerned whether the perturbation introduced generalizes to the diverse time-series forecasting domains. Even within the financial domain, it is unclear what the motivation is behind the designed perturbations.
- The multimodality illustrated in the paper is fairly limited. In the financial domain, real financial multimodality is substantially more diverse and may include order-book data, reports, signal indicators, news, analyst reports, and market sentiment. These introduce additional noise that can affect robustness, such as missing modalities, asynchronous observations, and contradictory correspondences.
- Consequently, the evaluated multimodal models are limited, and robustness is not fully justified. The experiments show robustness for particular visual transformations of price data under particular perturbations, rather than robustness across heterogeneous financial modalities.
- The paper explores a relatively narrow setting of financial data, which makes stating the framework for assessing general time-series foundation models overclaiming.

**Additional Comments:**

N/A

**Audience:**

Yes

**Audience Explanation:**

The paper addresses model robustness, causal evaluation, foundation models, and time-series forecasting, which are relevant to parts of the TMLR audience.

**Claims And Evidence:**

Yes

**Claims Explanation:**

Partially. Although the paper provides substantial experimental evidence that the proposed framework can distinguish model behavior under the six selected perturbations, the broader claims regarding financial time-series robustness and the superiority of multimodal forecasting systems are only partially supported.

**Requested Changes:**

Most of the requested changes match the weakness above. Below are some recommended changes suggested from most critical to least critical:

1. The authors should discuss and include (if possible) robustness considering realistic temporal distribution shifts.
2. The scope of the robustness claims should be better clarified.
3. The multimodal claims should be strengthened or reworded considering the diverse, heterogeneous modalities presented in the field.
4. Provide stronger justification for the selected perturbations.
5. Expand the financial datasets and forecasting settings.

---

### Review · Reviewer_ewhL · 2026-07-21

**Summary Of Contributions:**

The paper proposes a causally grounded rating framework that evaluates robustness and fairness of time-series forecasting models under noisy or erroneous inputs.

## Strengths
- The paper presents a clear motivation and addresses a relevant real-world problem.
- The work includes large-scale experiments and a comprehensive evaluation

## weaknesses
The manuscript requires substantial improvement in writing quality, structure, and academic presentation. Many sections need careful proofreading and revision to ensure clarity, consistency, and a professional academic style.

- The contribution section should be revised to better emphasise the novel contributions of the work, the specific challenges addressed, and how the proposed approach advances existing research. Currently, it mainly lists the components of the work and the experiments conducted, rather than clearly explaining the research gap, the novelty of the proposed solution, and its significance.

- The related work section needs restructuring. Currently, it primarily consists of a list of references with minimal discussion regarding their context and relevance to this study. The authors should offer a more critical analysis of existing approaches and explain how these works and methods relate to the problem addressed in this paper. For instance, various methods, tools, and models from different domains are mentioned, but the authors should clarify how these connect to the proposed work.

- The problem definition is not sufficiently clear. Several terms, variables, and definitions are introduced without providing adequate context regarding their relationships, their role in the proposed contribution, or to the specific problem being addressed. The authors should clearly define the problem and establish a clear connection between the problem formulation and the proposed solution. For example, you mentioned the aim is to highlight the worst-case scenario. The authors should explain what this scenario represents

- The proposed solution approach requires further clarification. The relationship between the proposed method, the problem definition, and the evaluation criteria is not sufficiently explained. The authors should provide a clearer description of the methodology.

- The workflow presented in Figure 3 is not entirely clear. Specifically, it is difficult to understand how the final prediction is generated. The figure appears to suggest that the predictions from all models are combined.

- The evaluation could be strengthened by including comparisons with more recent and advanced deep learning models for time series forecasting. The current evaluation relies heavily on general-purpose models, whereas incorporating state-of-the-art forecasting models would provide a more comprehensive assessment

**Audience:**

Yes

**Audience Explanation:**

The paper is relevant to the scope of TMLR, as it addresses topics in AI robustness, machine learning evaluation methodologies, and responsible AI for real-world time series forecasting models

**Claims And Evidence:**

Yes

**Claims Explanation:**

Yes. The claims are generally supported by comprehensive experimental evidence. The authors conduct large-scale experiments across six perturbation settings and evaluate the proposed approach on 11 models across multiple datasets.

**Requested Changes:**

- Improve the overall writing quality and academic presentation

- Revise the contribution section

- Strengthen the related work discussion
 - Clarify the problem formulation
- Clarify the proposed methodology and solution framework

- Expand the baseline comparison

For details, please check my comment above